# Sweat permeable and ultrahigh strength 3D PVDF piezoelectric nanoyarn fabric strain sensor

Wei Fan [1] ✉, Ruixin Lei[1], Hao Dou[1], Zheng Wu [1], Linlin Lu[1], Shujuan Wang[2], Xuqing Liu[3], Weichun Chen[1], Mashallah Rezakazemi [4] ✉, Tejraj M. Aminabhavi [5], Yi Li [6] & Shengbo Ge [7] ✉

Commercial wearable piezoelectric sensors possess excellent anti-interference stability due to their electronic packaging. However, this packaging renders them barely breathable and compromises human comfort. To address this issue, we develop a PVDF piezoelectric nanoyarns with an ultrahigh strength of 313.3 MPa, weaving them with different yarns to form three-dimensional piezoelectric fabric (3DPF) sensor using the advanced 3D textile technology. The tensile strength (46.0 MPa) of 3DPF exhibits the highest among the reported flexible piezoelectric sensors. The 3DPF features anti-gravity unidirectional liquid transport that allows sweat to move from the inner layer near to the skin to the outer layer in 4 s, resulting in a comfortable and dry environment for the user. It should be noted that sweating does not weaken the piezoelectric properties of 3DPF, but rather enhances. Additionally, the durability and comfortability of 3DPF are similar to those of the commercial cotton T-shirts. This work provides a strategy for developing comfortable flexible wearable electronic devices.

Flexible piezoelectric sensors have widespread applications in detecting physical signs and body posture, as they can convert mechanical signals into electrical signals for transmitting and processing information[1–5]. Examples of applications include real-time monitoring of arterial pulse[6], human movement[7–9], continuous blood pressure[10,11], and human-machine interaction[12]. However, the existing flexible piezoelectric sensors are packaged with impermeable membranes[13,14], thereby reducing the ambient moisture impact on device sensing performance and preventing short-circuiting[15]. As a result, the user may experience discomfort due to interference with the circulation and exchange of gases between the body and the external environment[16]. In addition, if sweat is not cleaned up for a prolonged time, it can accumulate and cause inflammation as well as other skin diseases[17,18].

Nanofiber membrane-based piezoelectric sensors possess good breathability[19]. Poly(vinylidene fluoride) (PVDF) has been the most commonly used piezoelectric polymer, which can be directly polarized to form piezoelectric membranes through electrostatic spinning[20,21]. Feng et al. stitched a PVDF nanofiber membrane as a piezoelectric layer and two conductive fabrics as an electrode layer to form a

[1]School of Textile Science and Engineering, Key Laboratory of Functional Textile Material and Product of Ministry of Education, Institute of Flexible electronics and Intelligent Textile, Xi'an Polytechnic University, Xi'an, Shaanxi, China. [2]School of Chemistry, Xi'an Jiaotong University, Xi'an, China. [3]State Key Laboratory of Solidification Processing, Center of Advanced Lubrication and Seal Materials, School of Materials Science and Engineering, Northwestern Polytechnical University, Xi'an, China. [4]Faculty of Chemical and Materials Engineering, Shahrood University of Technology, Shahrood, Iran. [5]Center for Energy and Environment, School of Advanced Sciences, KLE Technological University, Hubballi, India and Korea University, Seoul, Republic of Korea. [6]Department of Materials, University of Manchester, Oxford Road, Manchester, UK. [7]Co-Innovation Center of Efficient Processing and Utilization of Forest Resources, College of Materials Science and Engineering, Nanjing Forestry University, Nanjing, China. ✉e-mail: fanwei@xpu.edu.cn; mashalah.rezakazemi@gmail.com; geshengbo@njfu.edu.cn

piezoelectric sensor[22]. The sensor is breathable, but the conductive fabric cannot apply directly to the skin, where it must be packaged to create an airtight device. Fan et al. stitched Janus conductive fabrics and PVDF nanofiber membranes together to form the piezoelectric devices[23,24]. The conductive side of the Janus conductive fabrics is close to the piezoelectric layer, while the non-conductive side can be directly adjacent to the skin, which solves the problem of poor permeability of piezoelectric sensors. However, piezoelectric sensors based on nanofiber membranes cannot withstand high mechanical demands for a long time since nanofiber membranes have poor mechanical properties[25].

Yarn-based piezoelectric sensors, by contrast are breathable and are able to withstand greater strength[21]. Yang et al. woven PVDF-TrFE nanoyarn, conductive yarn, and non-conductive yarn into two-dimensional woven piezoelectric fabric[26]. Soin et al. knitted a three-dimensional (3D) spacer piezoelectric fabric with a silver-coated poly-amide composite filament layer as the top and bottom electrodes and PVDF monofilament as spacer yarn[27]. Although the above two piezoelectric fabrics have strong mechanical and piezoelectric properties, their outermost layer is still the conductive fabric that requires further packaging for practical use, thus affecting the air permeability of the device. Ahn et al. used a non-conductive 3D spacer fabric to encapsulate PVDF piezoelectric film coated with a conductive layer on both sides to form a breathable 3D piezoelectric sensor[28]. However, the multi-layered structure of the resulting sensor prevents sweat from being removed from the skin in time and subsequently affects the comfort of wearing. Fortunately, the 3D orthogonal fabric is highly designable[29,30]. The fabric has multiple layers of X, Y, and Z-direction yarns that are perpendicular to each other, allowing it with an asymmetric wetting structure by laying out yarns with different hygroscopic properties in various layers[31]. Through the structure design, the woven fabric has the function of one-way liquid transport, which can automatically transfer sweat from the hydrophobic layer to the hydrophilic layer on the far side of the skin, hence ensuring the comfort to the human body.

In the structure of the piezoelectric nanogenerator, the electrode is responsible for transferring electrical energy generated by the piezoelectric material to other areas, such as load or electric consumers. Therefore, the selection, design, and manufacture of the electrode materials are of utmost importance[32]. The commonly used electrodes include metal-based electrodes[33], carbon-based, ink-based, conductive polymer-based, and conductive threads. Since the electrode in this work needs to be woven in fabric, silver-plated nylon yarn is an optimal selection due to its exceptional electrical conductivity, high strength, and remarkable flexibility[34]. In this research, we developed a 3D piezoelectric fabric (3DPF) sensor with unidirectional water transport by reasonably designing the yarns (PVDF nanoyarns, silver-nylon yarns, Coolmax yarns, viscose yarns, and polyester yarns) with different hygroscopic properties (Fig. 1). The PVDF nanoyarn possesses piezoelectric properties, which enable it to generate an electrical signal under pressure. The silver-nylon yarn is responsible for transmitting the electrical signals produced by the PVDF fibers. Coolmax yarn, which has a profiled cross-section and excellent moisture conductivity, served as the medium for water transport. Viscose yarn composed of cellulose fibers with strong hygroscopic properties was used to absorb water. Polyester yarn with high strength and water absorption capabilities was employed to provide strength and moisture transport. Firstly, the high-performance PVDF nanoyarns were prepared by conjugate electrospinning and hot stretching. Subsequently, the yarns were woven using the fully automated rapier loom to form 3DPF with an orthogonal structure. We then evaluated 3DPF's wearing durability, comfort, and practical properties, such as piezoelectric signal resistance to sweat disturbance. Additionally, the hardware and software systems were designed with STM32 as the main control unit, which enabled the wireless transmission of piezoelectric signals by the 3DPF. Finally, 3DPF was explored as a self-powered switch for bed sheets to enable intelligent healthcare and for children's belts to provide alarm functions.

## Results

### Structure and properties of PVDF nanoyarns before and after hot stretching

PVDF piezoelectric nanoyarns were directly twisted from PVDF nanofibers polarized by high voltage in the conjugate electrospinning process. The speed at which nanofibers are wound has a direct impact on their orientation and subsequently the mechanical properties of nanoyarns[35]. When the collection winding speed is 1.0 mm s⁻¹, the

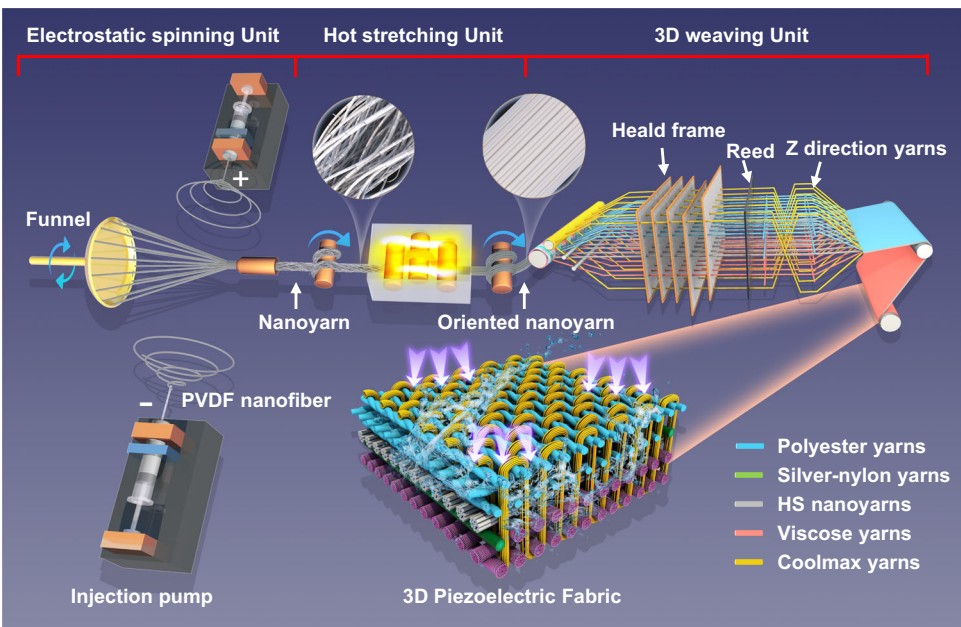

**Fig. 1 | Manufacturing and structure schematic diagram of three-dimensional piezoelectric fabric (3DPF).** Ultrahigh strength PVDF piezoelectric nanoyarns were obtained by conjugate electrospinning (Electrostatic spinning unit) and hot stretching (Hot stretching unit). They were then combined with functional yarns to weave the 3DPF strain sensor on a fully automatic rapier loom (3D weaving unit).

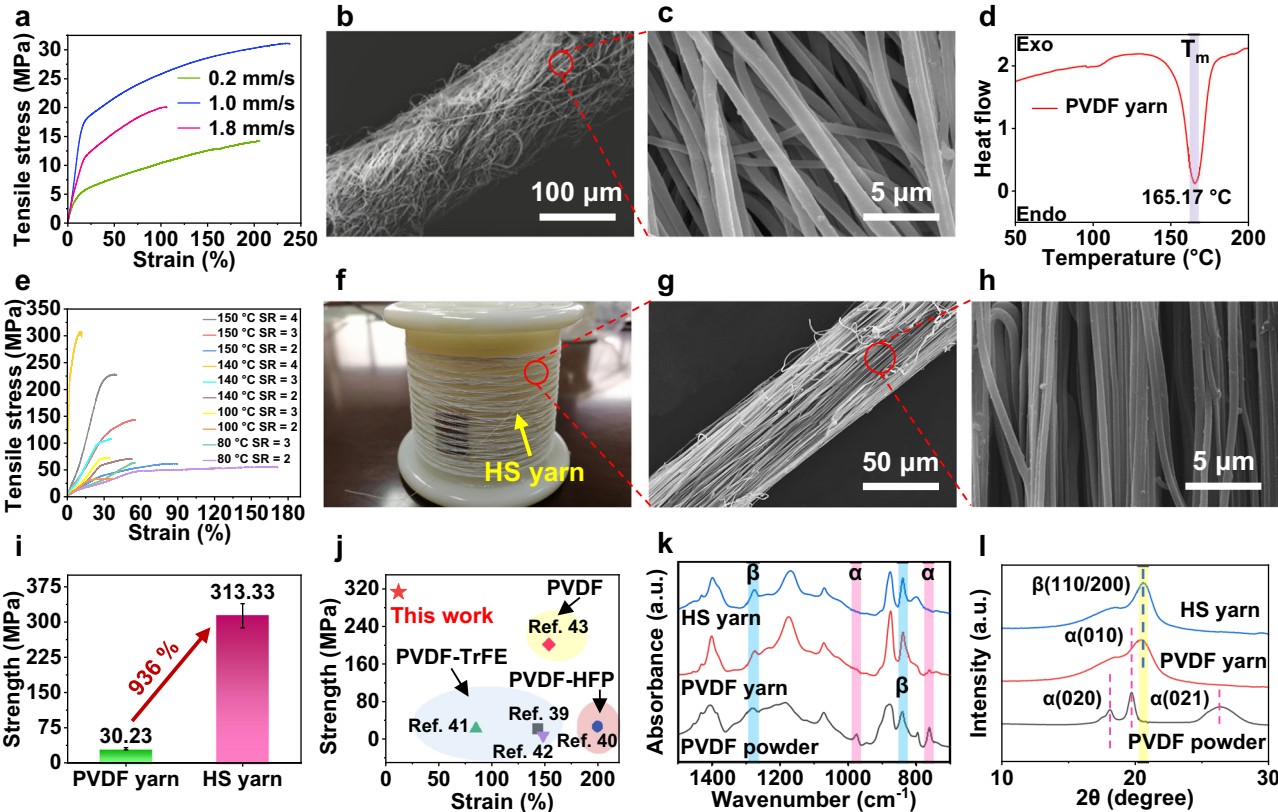

**Fig. 2 | Morphology and properties of PVDF nanoyarns before and after hot stretching. a** Stress-strain curves of PVDF as-spun nanoyarn at different collection speeds. **b**, **c** SEM image of PVDF as-spun nanoyarn. **d** Differential scanning calorimetry (DSC) data of PVDF as-spun nanoyarn. **e** Stress-strain curves of hot stretching nanoyarns (HS yarn) at different hot stretching conditions. **f** Photograph and **g**-**h** SEM images of HS yarn. **i** Maximum strength of PVDF nanoyarn and HS yarn. **j** Tensile strength comparison between HS yarn and the other PVDF and its copolymers nanoyarns. **k** FTIR and **l** XRD patterns of the HS yarn, PVDF nanoyarn and PVDF powder, respectively.

strength of the as-spun PVDF nanoyarns is 30.2 ± 2.2 MPa (Fig. 2a). It is worth noting that while the strength of the nanoyarn is currently at its peak, it is somewhat fluffy and contains many disorganized nanofibers. Additionally, the diameter of the nanoyarn and the nanofiber is around 109 μm and 850 nm, respectively (Fig. 2b, c).

Hot stretching can enhance the mechanical properties of PVDF nanoyarn, and improve the β-phase content, which is positively correlated with its piezoelectric properties[36,37]. The hot stretching temperature should be chosen between the glass transition temperature ($T_g$) and the melting temperature ($T_m$) of the polymer[37]. The $T_m$ of the as-spun PVDF nanoyarns is about 165 °C (Fig. 2d), whereas the $T_g$ of PVDF is about −35 °C[38]. Therefore, the hot stretching temperatures were set at 80, 100, 140, and 150 °C to obtain the hot stretching nanoyarns (HS yarn). It was found that the strength of HS yarns at 140 °C is higher than HS yarns obtained at other temperatures under the same stretching ratio (SR represents the length ratio of the yarn after and before stretching) (Fig. 2e). Before 140 °C, HS yarn mechanical properties positively correlate with increasing temperature. However, once the temperature surpasses 150 °C, the tensile strength of HS yarn becomes lower than that at 140 °C. This phenomenon occurs due to the intensified thermal motion of the macromolecular chains in the fiber, leading to a relative slip and a subsequent reduction in breaking strength. Hence, it can be concluded that the optimal stretching condition for HS yarn is 140 °C. Besides, the strength of the HS yarn is proportional to SR at the same stretching temperature. The HS yarn strength reaches the maximum value of 313.3 ± 15.8 MPa at the stretching conditions of 140 °C and SR = 4. This is because the greater SR leads to a higher orientation and a smaller diameter of the nanofiber (525 nm) in the yarn (54 μm) (Fig. 2f-h), thus

improving the strength of the HS yarn. The strength of the HS yarn obtained under the optimum hot drafting condition is about 936% better than that of the as-spun PVDF nanoyarn (Fig. 2i) and higher than that of all PVDF and its copolymers nanoyarns so far (Fig. 2j)[39–43]. Our research has shown that PVDF nanoyarns without hot stretching are unsuitable for 3D automatic looms. This is because the fibers on the surface of the PVDF nanoyarns (Fig. 2b) have less axial orientation, and the hairiness is serious after repeated friction with the steel buckle during the weaving process. This hairiness leads to entanglement between yarns which hinders continuous weaving. The mechanical properties of the HS yarn are high enough to meet the requirements of weaving on 3D automatic looms (Supplementary Movie 1).

PVDF powder has five crystal forms: α, β, γ, δ, and ε-phase, among which the most common crystal type is the nonpolar α-phase[44]. The α-crystal chain dipole is the opposite and does not show polarity, but it can be transformed into other phases, such as β-phase, under sufficient mechanical stress, heat, or electricity[45,46]. The β-crystal cell contains polar zig-zag chains, which is the key to the piezoelectric properties of PVDF. The characteristic peaks at 764 and 975 cm⁻¹ wavenumbers are α-phase, while those at 841 and 1275 cm⁻¹ wavenumbers are β-phase[47]. In this work, most of the α-phase was polarized to β-phase in the process of high voltage electrostatic spinning and hot stretching process (Fig. 2k). Therefore, characteristic peaks at 764 and 975 cm⁻¹ of PVDF and HS yarn decrease, while those at 841 and 1275 cm⁻¹ increase. The β-phase content can be calculated by the Beer-Lambert law[48]:

$$F(\beta) = \frac{A_\beta}{\left(\frac{K_\beta}{K_\alpha}\right)A_\alpha + A_\beta} = \frac{A_\beta}{1.26A_\alpha + A_\beta} \quad (1)$$

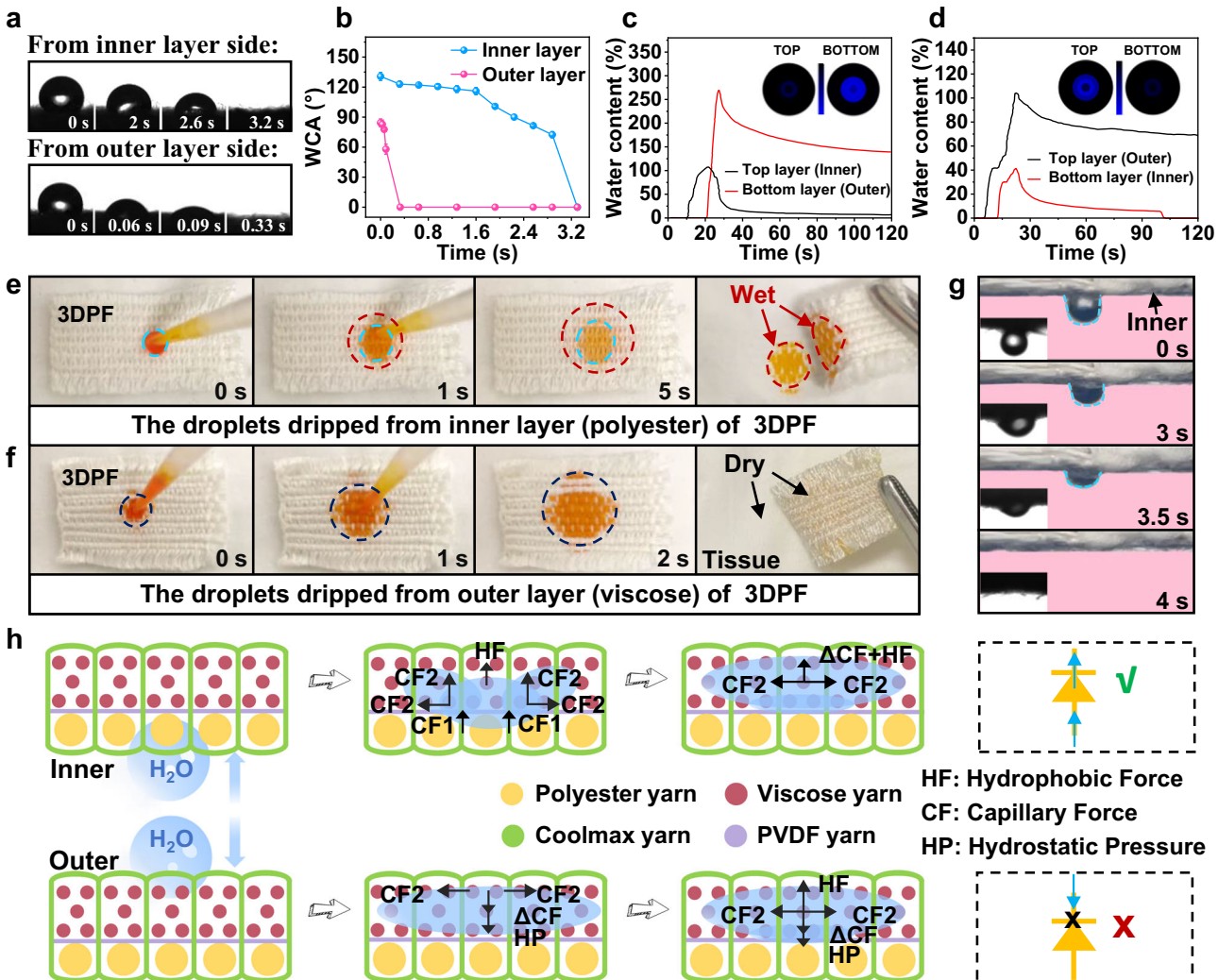

**Fig. 3 | Unidirectional water transport property of 3DPF.** Optical microscopic images of **a** water droplets and **b** dynamic water contact angles (WCA) on both sides of 3DPF at different times. Results of the water management tester (MMT) for 3DPF when **c** the inner layer up and **d** the outer layer up. 30 μL orange ink was dripped on the **e** inner and **f** outer layers of the 3DPF, showing the unidirectional liquid transport capacity. **g** The anti-gravity experiment of the 3DPF. **h** Diagrams showing the simplified unidirectional water transport mechanism of the 3DPF.

where, $A_\alpha$ and $A_\beta$ are absorbance at wavenumber 766 and 841 cm⁻¹; $K_\alpha$ and $K_\beta$ represent the absorption coefficients at corresponding wave numbers of each crystal phase, which are $6.1 \times 10^4$ and $7.7 \times 10^4$ cm² mol⁻¹, respectively. The results indicate that the β-phase levels in powder, as-spun yarn and HS yarn are 54.34, 85.88 and 88.08%, respectively. The crystal phase change of the PVDF powder, PVDF nanoyarns and HS yarn in the XRD pattern proves the existence of the above phenomenon again (Fig. 2l). For the PVDF powder, the diffraction peaks detected at $2\theta = 18.1°$, 19.7°, and 26.4° correspond to the (020), (110), and (021) crystal planes of α-phase PVDF, respectively[49]. For the yarns, the diffraction peaks at $2\theta = 20.8°$ correspond to the (110) and (200) crystal planes of β-phase[26,50]. The peak strength and peak area of HS yarn are stronger than that of PVDF nanoyarns, suggesting that hot stretching promotes the formation of β-phase and causes the HS yarn to have higher grain plane orientation and crystallinity.

**Unidirectional water transport property of 3DPF**

The 3DPF has multiple layers of structure in which the inner layer made of polyester yarns is near the body side, and the outer layer made of viscose yarns is away from the skin. Figures 3a and b display the captured optical image and apparent water contact angle (WCA) changes

as water drops from the inner and outer layers of the 3DPF, respectively. The WCA of the inner layer is maintained at about 120° for 1.5 s. After that, the water absorption rate becomes faster and the droplets are completely absorbed in 3.2 s. In the initial state, the outer layer has a contact angle of 87.8°, and the water droplet on the outer layer is completely absorbed after only 0.33 s. These phenomena indicate that the 3DPF has a hydrophobic inner layer and a hydrophilic outer layer.

The structure of 3DPF formed a wettability gradient along the Z yarn direction. A moisture management tester (MMT) was used to measure the wetting performance of the 3DPF quantitatively. Figures 3c and d show the changes in relative water content on both sides when artificial sweat is continuously added vertically to the upper surface of the 3DPF sample, with the inner layer up (Fig. 3c) and the outer layer up (Fig. 3d). When selecting the inner layer as the upper surface, the water content on the top layer eventually maintains near zero, while the water content at the bottom layer increased gradually and reached its peak (140%) at 26 s, indicating that the moisture in the inner layer of 3DPF has been transferred to the outer layer. When selecting the outer layer as the upper surface, the water content of the outer layer is always higher than that of the inner layer. As a result, the inner layer remained completely devoid of water, inferring that the sweat of the hydrophilic outer layer was almost blocked by the inner

layer. Overall, the findings suggest that the 3DPF has a one-way water transport capacity. To better comprehend the unidirectional liquid transport performance of the 3DPF, 30 μL orange ink was dropped on the inner layer. It was observed that the ink rapidly penetrated from the inner layer to the outer layer and diffused in the outer layer, finally wetting the beneath napkin (Fig. 3e). Conversely, when the ink was dropped on the outer layer, the water spread quickly on the outer layer without penetrating the inner layer (Fig. 3f). The water is transported in one direction (inner layer to outer layer) and not in the opposite direction. Then, the anti-gravity experiment (Supplementary Movie 2) was performed on the 3DPF, as shown in Fig. 3g. The droplet travels from the bottom side (inner layer) to the top side (outer layer) of the 3DPF in 4 s. The results show that the 3DPF has anti-gravity unidirectional water transport performance that helps keep the skin dry and comfortable by transporting sweat from the inner layer near the skin to the outer layer in time.

The superior unidirectional liquid transportation ability of 3DPF is attributed to its inherent asymmetrical wettability and the strong core suction ability of the Z-oriented yarns with irregular sections. The water absorption of viscose yarn is 708.2 %, which is more than twice that of polyester yarn (298.6%). While Coolmax yarn has a lower absorption rate than viscose, it still exceeds polyester fibers. On the other hand, PVDF yarn exhibits a much lower water absorption rate (69.4 %) than viscose and polyester yarn (Supplementary Fig. 1). The Coolmax yarns in 3DPF are oriented in a Z-direction and positioned perpendicular to the fabric surface[31]. This unique configuration results in a strong core suction effect (Supplementary Fig. 2) due to the multi-grooved surface structure of the Coolmax yarn (Supplementary Fig. 3).

The findings revealed that polyester yarn and PVDF nanoyarn exhibit hydrophobic characteristics with water contact angles of 130.7° and 119°, respectively (Supplementary Fig. 4a and 4b). Conversely, viscose yarn possesses hydrophilic properties with a WCA of 87.8° (Supplementary Fig. 4c). Furthermore, a gradient wetting effect is observed in the thickness direction of the 3DPF owing to the increasing hydrophilic performance from the polyester layer to the PVDF layer, followed by the viscose layer. The unidirectional liquid transport mechanism of the 3DPF is shown in Fig. 3h. When water is dropped on the polyester yarns side of the 3DPF, it follows the path of the Z-oriented yarns and overcomes gravity. This is because the Z-oriented yarns made of Coolmax material possess greater core absorption and water absorption capacity (force CF1) than polyester yarns. The strong capillary force of Z-oriented yarns propels the droplets upward. The water conveyed by the Z-oriented yarns from the bottom passes through the PVDF yarn layer, which has weak water absorption and is subsequently absorbed by the viscose yarns on the upper layer of the 3DPF due to their super water absorption capacity (force CF2). Upon touching the upper layer, the droplet experiences an upward hydrophobic force (HF) imparted by the inner layer. Additionally, there is also a capillary force CF2 on the upper layer in the vertical and horizontal direction, which causes the droplet to spread laterally and subsequently wet the upper layer. The droplet is transported to the upper layer without reverse transport, keeping the inner layer dry. The anti-gravity unidirectional liquid transport phenomenon of the 3DPF has been investigated numerically using the COMSOL Multi-physics simulation software, and the simulation results agree with the experimental results (Supplementary Fig. 5, Supplementary Movie 3 and Supplementary Movie 4). On the contrary, when the droplet drops on the outer layer, it enters the outer layer and rapidly diffuses laterally because of the hydrostatic pressure (HP) and ΔCF (the sum of the capillary force CF1 of the Z yarn and CF2 of the outer layer). Now, the net force on the drop is pointing straight down. However, the droplet stops falling when it reaches the inner layer, where the inner layer provides upward hydrophobic force HF and prevents the droplet from moving further down.

## Piezoelectric properties of the 3DPF

HS nanoyarns were used as a piezoelectric layer in the 3DPF. The piezoelectric mechanism diagram of the 3DPF is reflected in Fig. 4a. Electrical signals generated by the PVDF piezoelectric layer are derived from electrically conducting yarns above and below the piezoelectric layer in 3DPF.

The 3DPF is a sensor without packaging, so whether human sweating will affect its electrical signal output is the key to be investigated in this paper. Figure 4b and c illustrate the output piezoelectric signals of the 3DPF and the 3DPF containing sweat at a frequency of 2.5 Hz, respectively. For the 3DPF, under the applied pressure of 0.5 to 20.0 N (1.25 to 50 kPa), the output voltage generated by the 3DPF gradually increases from 0.51 to 1.02 V, and the output current continuously increases from 28.80 to 75.60 nA (Fig. 4b). The output signal of the 3DPF at different frequencies is positively correlated with the frequency (Supplementary Fig. 6). In addition, the output signals of the 3DPF with 10 μL artificial sweat were tested (Supplementary Movie 5) to simulate the piezoelectric properties of the 3DPF after absorbing human sweat. The output voltage of the 3DPF containing sweat increases from 4.94 V at 0.5 N to 8.45 V at 8 N, then decreases to 4.40 V at 20 N. The corresponding output current increases from 513 to 1080 nA and then decreases to 520 nA (Fig. 4c).

The output signal of 3DPF containing sweat is positively related to the force in the range of 0.5–8.0 N, and its output voltage and current are about 10 and 20 times larger than those of the 3DPF, respectively (Fig. 4b, c). When the sweat is evaporated, the output signal of the dry 3DPF is almost the same as the original 3DPF (Supplementary Fig. 7). As a result of human sweat, the piezoelectric properties of the 3DPF are not weakened but are enhanced. This is because suitable moisture amount helps to increase ferro-/piezoelectric performance of the piezoelectric materials[51–55]. In addition, all the piezoelectric materials are dielectric and present an insulating state[56]. Piezoelectric materials can improve their piezoelectric properties by introducing the defects[57,58]. The sweat is a kind of charge carrier, and hence, the insulated PVDF yarn could have been semi-conducted by the introduction of sweat defects, which in turn enhances the piezoelectric properties of 3DPF.

The reason for the decrease of the generated piezoelectric signal of 3DPF containing sweat at 20 N may be related to the theoretical limit of the effective strain of piezoelectric material in high-pressure regions[48,59,60]. It is possible that sweat can reduce the resilience of the 3DPF, and excessive load causes the 3DPF strain recovery to slow down. Namely, the strain has not returned to zero before entering the next strain, resulting in a relatively small strain in the material when the pressure is applied in the subsequent piezoelectric cycle test. As a result, the output voltage decreases.

The sensitivity and response time of 3DPF are the key factors to its application as a sensor. The sensitivity is defined as the ratio between the output change (ΔU) and the input change (ΔP) of the sensor under steady working conditions, which reflects the measurement accuracy of the sensor. The formula of sensitivity is as follows[12]:

$$S_\upsilon = \frac{\Delta U}{\Delta P} \ (\upsilon = 1,2) \tag{2}$$

where $\upsilon$ represents the constants. According to Eq. (2), at low-pressure range of 0–1.25 kPa, the voltage sensitivity values of 3DPF, before and after sweating are 0.41 and 3.95 V kPa$^{-1}$, respectively (Fig. 4d and Supplementary Fig. 8). The response time of the sensor is defined when the signal rises from its initial state to 90% of its peak value[21]. As illustrated in Fig. 4e, the response time of the 3DPF decreases from 100 ms in the dry state to 50 ms in the wet state (with 20 μL sweating) and reduces to ~24 ms as the amount of sweat was increased to 100 μL, as shown in Supplementary Fig. 9. While 3DPF has a qualified sensitivity in the dry state, it achieves the maximum value in the wet state when

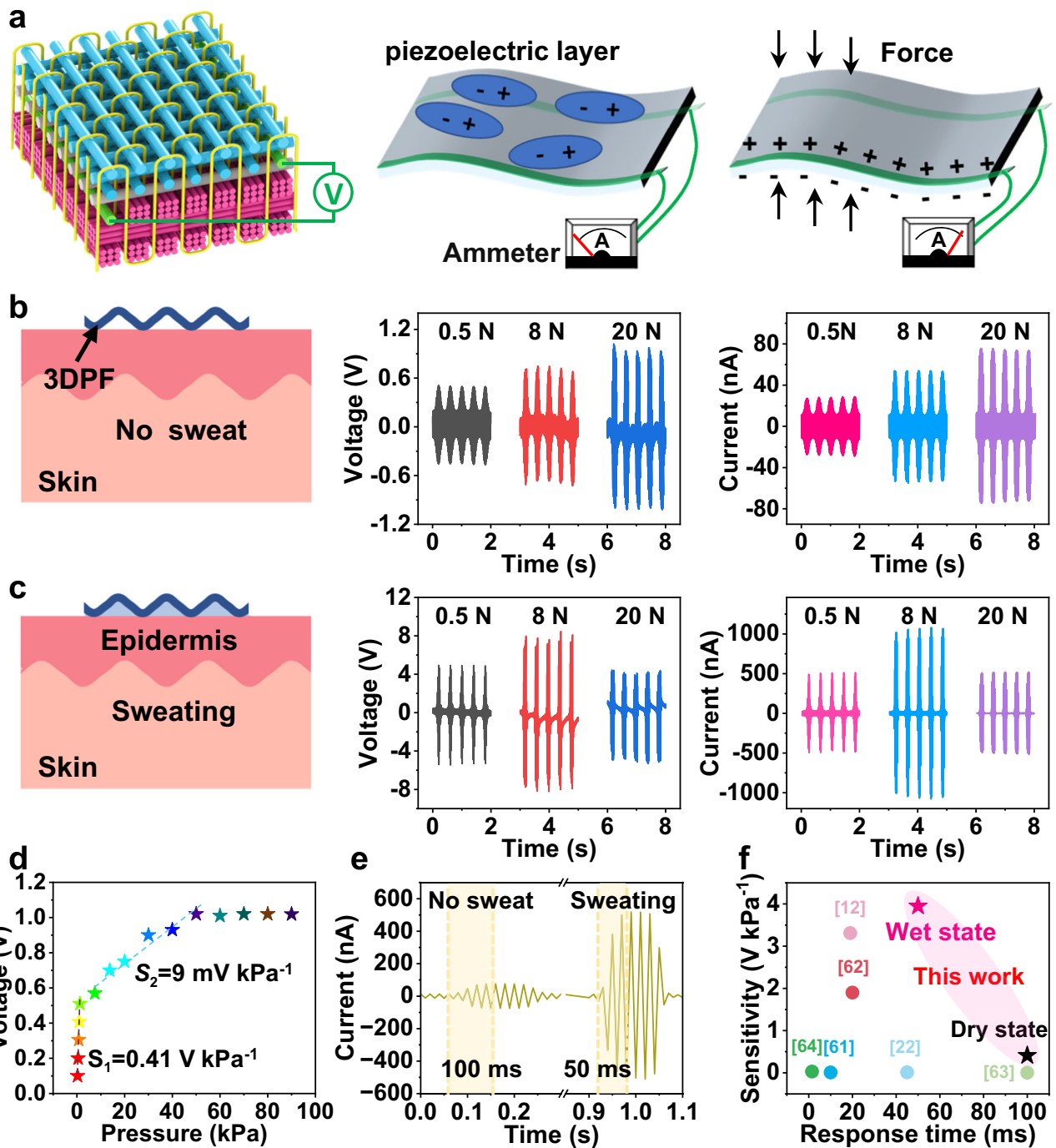

**Fig. 4 | Piezoelectric properties of the 3DPF. a** The piezoelectric mechanism diagram of the 3DPF. Piezoelectric outputs of the 3DPF **b** before and **c** after sweating. **d** Voltage sensitivity and **e** response time of the 3DPF before and after sweating. **f** The sensitivity of 3DPF compared with other flexible piezoelectric sensors.

compared to other representative flexible piezoelectric sensors (Fig. 4f)[12,22,61–64]. Owing to the special structure of 3DPF, the sensitivity of 3DPF is higher than most other PVDF-based piezoelectric sensors[65–70] (Supplementary Table 1). The high sensitivity and the fast response time of the 3DPF containing sweating ensure its effectiveness as a piezoelectric sensor in actual use such as human sweating conditions.

### Durability and comfort of the 3DPF

The tensile strength of 3DPF is 46.0 ± 4.3 MPa (Fig. 5a), which is the strongest as compared to those flexible piezoelectric pressure sensors ever reported (Fig. 5b)[7,71–75]. The inner and outer layers of 3DPF exhibit

wear resistance times of approximately 32,000 and 19,000, respectively, which is slightly lower than that of commercial denim yet surpasses that of commercial cotton fabric (Fig. 5c). All in all, the wear resistance of the inner and outer layers of 3DPF satisfy the standard requirements of more than 10,000 times in the GB/T 21295-2014 (the technical standards of physical and chemical properties of clothing). A slight decrease of ~3% in the output voltage of the 3DPF is detected for over 20,000 cycles under 20 kPa and frequency of 4 Hz, demonstrating the excellent mechanical durability of the 3DPF (Fig. 5d). Moreover, the property of being washable is a fundamental requirement for fabrics. The output voltage of 3DPF was tested after various washing cycles. As shown in Supplementary Fig. 10, the output voltage of 3DPF

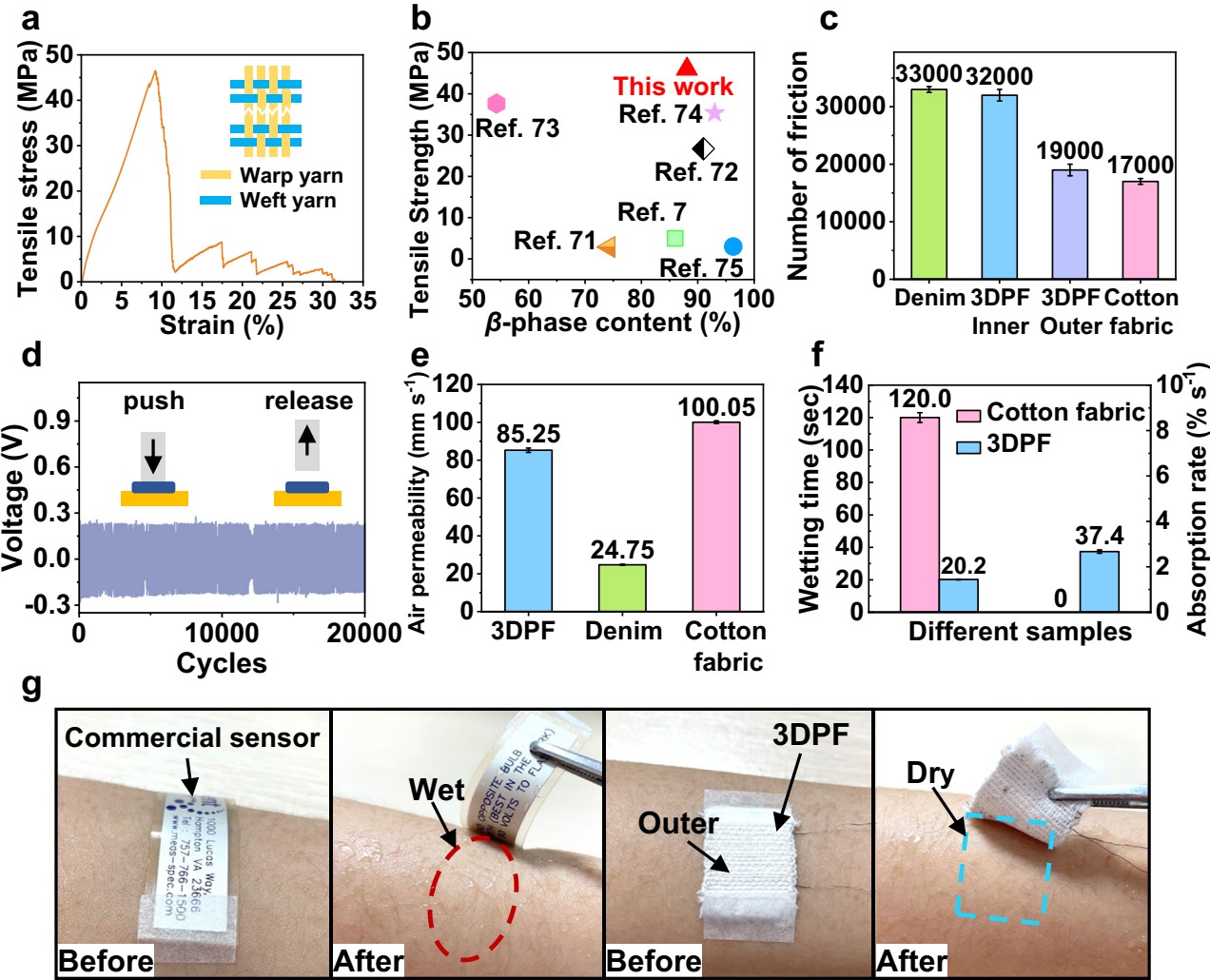

**Fig. 5 | Durability and comfort of the 3DPF. a** Tensile stress-strain curve of the 3DPF. **b** Comparison of tensile strength of the 3DPF with the other flexible piezoelectric pressure sensors. **c** Comparison of wear resistance of the 3DPF with denim and cotton fabric. **d** Durability of the 3DPF. **e** Comparison of air permeability of the 3DPF with denim and cotton fabric. **f** Initial wetting time and absorption rate at the bottom of cotton fabric and the 3DPF. **g** Comparison of the accumulation of sweat on the skin surface for commercial piezoelectric sensor and the 3DPF after human sweating.

shows no significant decrease after ten washing cycles. The results suggest that 3DPF possesses excellent washability.

The air permeability of 3DPF is 85.25 mm s⁻¹, which is smaller than commercial cotton fabric (100.05 mm s⁻¹) but higher than denim (24.75 mm s⁻¹) (Fig. 5e). The wetting time of the 3DPF is 20.2 s, which is only 20% of the time recorded for the cotton fabric in T-shirt (120 s) (Fig. 5f). The absorption rate of the 3DPF is 37.4% s⁻¹, while that of the commercial cotton fabric is zero (Fig. 5f). These findings indicate that the hygroscopic and perspiring performance of the 3DPF is much better than that of the cotton fabric. Moreover, a commercial PVDF piezoelectric sensor and the 3DPF were pasted on the volunteers' left arm and right arm, respectively to test the actual wearing comfort performance under running. It is observed that when the human body sweats after exercising, there is significant sweat between the commercial PVDF piezoelectric sensor and the skin, while the skin under the 3DPF is dry due to its unidirectional water transport (Fig. 5g), therefore ensuring a comfortable experience for the wearer. Human sweat comprises various organic metabolites, including glucose, lactic acid, cortisol, and uric acid[76]. The odor is typically caused by the degradation of organic substances during the piezoelectric process[77]. After several sweating and drying cycles, no odor was detected in the experiment. Overall, the excellent durability and comfort properties of

the 3DPF allow it to be worn in the human body for an extended period.

## Applications of the 3DPF as a self-powered switch

The 3DPF has good wearability and sweat-enhanced electrical output performance that is suitable for long-term wear as a self-powered switch. As such, a trigger circuit was designed to realize the function of the 3DPF as a switch to trigger a functional device (Fig. 6a). The 3DPF was used as the piezoelectric sensor module, and the voltage was amplified and partitioned through the dual operational amplifier LMV358 and transmitted to the analog-to-digital converter of the Microcontroller Unit (MCU) for analysis and processing, and finally transmitted to the terminal. Among them, Arduino was selected as MCU, whose main role was to control the system power supply, detect the piezoelectric signal, and enable the level conversion chip.

Figure 6b shows the 3DPF acting as a trigger switch to drive the fan switch and indicator light on and off. When the 3DPF is pressed, the fan and the LED light immediately switch states and hold them until the next press (Supplementary Movie 6). On this basis, the clinical ward call system circuit was further developed. MCU uses STM32 to control Wi-Fi enablement and Universal Asynchronous Receiver/Transmitter (UART) communication (Supplementary Movie 7). The 3DPF can be

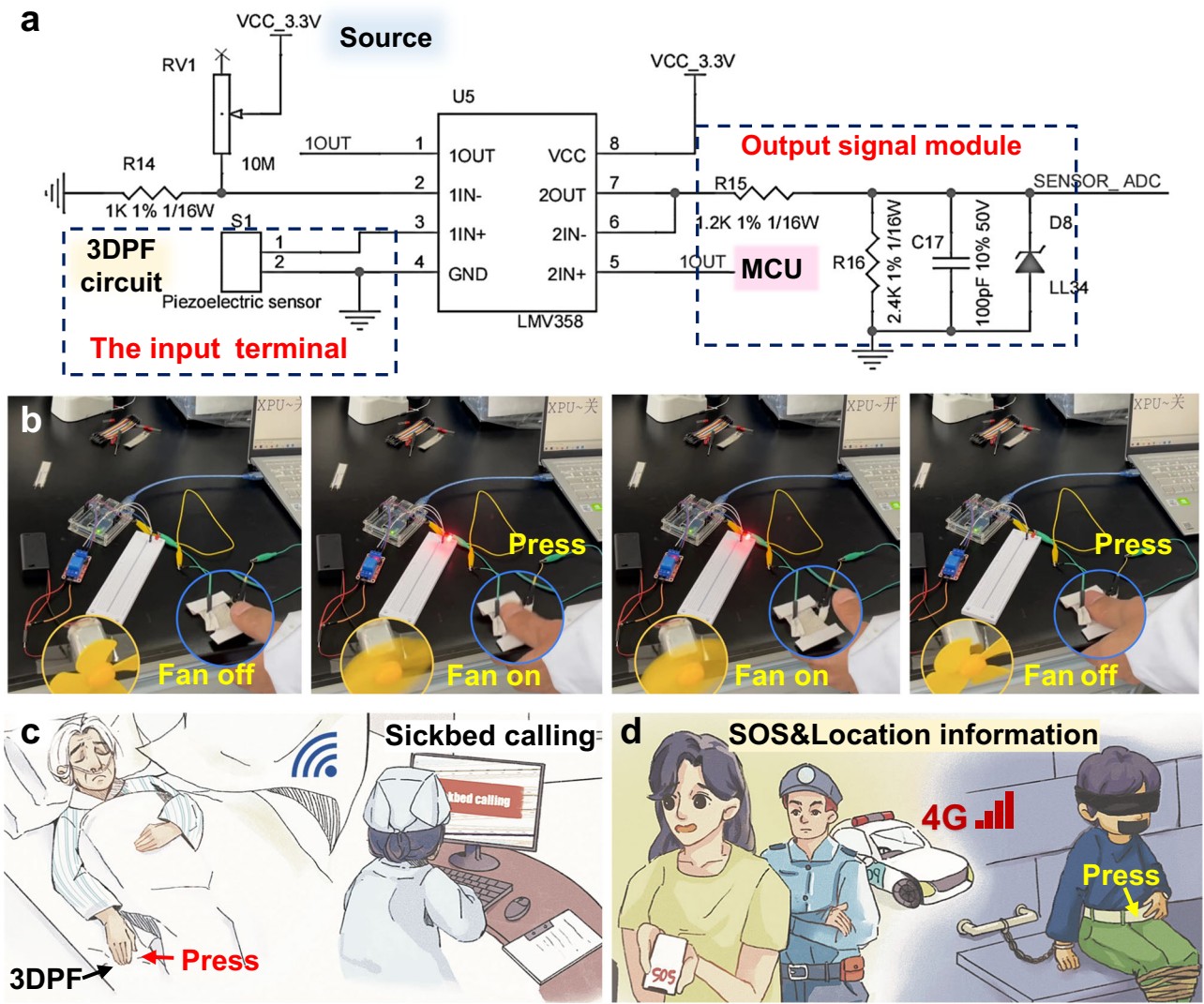

**Fig. 6 | The applications of the 3DPF. a** Schematic diagram of the 3DPF trigger circuit. **b** The small fan and LED light are triggered by pressing the 3DPF. **c** The 3DPF is used as a sheet with the function of sickbed calling. **d** The 3DPF is used as a belt for missing children's alarm and location.

utilized as a bedsheet with a trigger switch conveniently integrated on its edge. This feature enables patients to call for assistance without getting up by pressing the local trigger switch embedded in the sheet. The ward call signal can be transmitted through Wi-Fi to remote medical staff, ensuring timely care for patients. Additionally, the 3DPF sheet can achieve unidirectional fluid transport function, preventing bedsores, maintaining skin and sheet surface dryness, and ensuring patient comfort, especially in cases of urinary incontinence (Fig. 6c).

The piezoelectric yarn placed inside the 3DPF is discreet and resembles commercial fabric, making it an ideal component for creating a hidden switch on a 3DPF belt of a child to signal distress in case of an abduction (Fig. 6d). Unlike other wearable accessories such as watch bands or necklaces, 3DPF is less conspicuous and not easily removable by potential abductors. Unlike other wearable accessories such as watch bands or necklaces, 3DPF is less conspicuous and not easily removable by potential abductors. 4 G module (WH-GM5), GPS module (BD-126ZR), and other modules were used to develop the wireless alarm positioning system based on STM32 to realize the real-time transmission of signal and location information in the wide area network. The 4 G circuit mainly packages the product status information and piezoelectric sensor signals for external communication. Notably, the 3DPF belt is comfortable and undetectable. When the child is lost or in danger, the child can trigger the alarm switch by

pressing the belt. Then, the distress signal and real-time location information will be transmitted to the parent's device through 4G to facilitate timely rescue (Supplementary Movie 8).

## Discussion

Yarn is the smallest fabric unit with gaps between that allowing for good air permeability, enabling the gas exchange between the human body and the surroundings. In this study, exceptional PVDF piezoelectric nanoyarns with the highest strength (313.3 MPa) reported to date was successfully developed using conjugate electrospinning and hot stretching methods. The nanoyarns were then woven with common yarns of varying hygroscopic properties and conductive yarns to form a 3D piezoelectric fabric (3DPF) sensor using advanced 3D textile technology. The yarns within 3DPF are arranged based on their hydrophilic performance from lowest to highest. They are firmly held together by the Z-directional Coolmax yarns, which exhibit a strong core attraction effect. Since the yarns in 3DPF are straight, this particular component possesses the highest tensile strength among all the reported PVDF piezoelectric materials, reaching the value of 46.0 MPa. The 3DPF has an effective anti-gravity unidirectional water transport due to asymmetric wettability and capillary force of Z yarn. This allows the sweat to be transported from the inner layer near the human body to the outer layer in 4 s, thereby ensuring skin dryness and comfort. The PVDF

piezoelectric nanoyarns and conductive yarns are in the middle layer of 3DPF, while the inner and outer layers are composed of non-conductive polyester and viscose yarns. This eliminates the need for further sealing, providing comfort and durability like a commercial cotton T-shirt.

Furthermore, it is noteworthy that the piezoelectric properties of 3DPF are actually enhanced by human sweating rather than being weakened. Specifically, the 3DPF sensing sensitivity increases from 0.41 V kPa$^{-1}$ in the dry state to 3.95 V kPa$^{-1}$ in the wet state. In addition, the response time decreases from 100 ms in the dry state to 50 ms in the wet state. These remarkable properties render 3DPF a potential self-powered switch, which can be worn by individuals for extended periods, triggering alarm signals or location information and transmitting them via Wi-Fi or a 4 G module developed based on STM32, thus facilitating the timely rescue. The proposed intelligent wearable fabric strikes a balance between comfort and sensing properties, making it feasible for intelligent wearable products to be worn by the human body for prolonged periods.

## Methods

Materials: PVDF powder ($M_w$ = 1,100,000, Solvay, USA) was mixed in N, N-dimethylformamide (DMF, Aladdin Biochemical Technology, China) and acetone solution (Sinopharm Chemical Reagent, China) with a mass ratio of 7:3 followed by magnetically stirred at 70 °C for 12 h. The polyester yarns, silvered nylon yarns Coolmax yarns, and viscose yarns were purchased from Aobo Textile Co., Ltd (Shandong, China), Suzhou Xinwei Co., LTD (Jiangsu, China), Yueyi Textile Technology Co., Ltd (Zhejiang, China) and Tianpeng Textile Co., Ltd (Shandong, China), respectively. The orange ink was purchased from GRASP Stationery Co., Ltd, Zhejiang, China. The electronic components were purchased from Uxin Electronic Technology Co., Ltd (Shenzhen, China).

Preparation of PVDF nanoyarns and 3DPF: A conjugate electrostatic spinning device (Nayi Instrument, JDF05, China) was used to prepare PVDF as-spun nanoyarns. The spinning rate, voltage, and funnel speed were 0.6 mL h$^{-1}$, ± 5.3 kV and 200 rpm, respectively. The collecting roller speed was set as 1.0 mm s$^{-1}$. The spinning process is shown in Supplementary Movie 9. HS yarns were obtained by hot stretching, whereas the 3DPF was woven on a fully automatic rapier loom (SGA598, China). Detailed technological parameters of the yarns and the 3DPF are listed in Supplementary Table 2 and Supplementary Tab 3.

Characterizations: The surface morphology of the nanoyarns and nanofibers was observed by a scanning electron microscopic (SEM, Quanta-450-FEG + X-MAX50, Switzerland). The DSC curve was accessed from a differential scanning calorimeter (Q2000, TA). The crystal phase was analyzed by fourier transform infrared spectrometer (Nicolet iS50, USA) and X-ray diffraction analysis (Dmax-Rapid II). According to the ISO13934-2: 1999 standard, the tensile strength was assessed by the universal testing machine (UTM5205X, Shenzhen, China). The accuracy level of the machine is 0.5. The force control rate accuracy and deformation control rate accuracy are both within ± 1% of the set value when the rate is less than 0.05% FS s$^{-1}$. When the rate is greater than 0.05% FS s$^{-1}$, both are within ± 0.5% of the set value. Additionally, the precision of the displacement control rate is within ± 0.5% of the set value. The WCA was tested by a single-fiber contact angle meter (OCA40MICRO, Germany), and the accuracy of the video system is ± 0.1°. Moisture management performance was characterized by MMT (SDLATLAS, China) following AATCC 195-2011 standard.

A 3DPF sample with the size of 2 cm × 2 cm was placed on a force sensor (ZN5S-F, China), and the excitation was driven by a vibration generator (JZK-10, China) and a sweep signal generator (YE1311, China). The piezoelectric data was recorded by an electrometer (Keithley 6514, USA), which reads 1200 readings per second. The 3DPF was washed in a rigorous laundering environment according to AATCC standard 135. A sample with the size of 20 cm$^2$ was used for the gas permeability test (Automatic permeability meter, YG461E-III, China), the measurement accuracy is 1 mm s$^{-1}$. The wear resistance of the 3DPF sample was tested by a fabric flat grinder (YG401C-8, China) according to the ISO 12947-1:1998 standard. The technical standards of physical and chemical properties of clothing were performed according to the GB/T 21295-2014 standard. According to the standard protocol for repeated testing, at least five specimens were tested to ensure the consistency and repeatability of the results obtained.

## Data availability

The data that support the findings of this study are available in this article and supplementary materials. Additional data are available from the corresponding author upon request. Source data are provided with this paper.

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

## Acknowledgements

This work was partly supported by the National Natural Science Foundation of China (52073224, W.F.; 32201491, S.G.); Shaanxi Outstanding Youth Science Fund Project, China (2024JC-JCQN-03, W.F.); Young Elite Scientists Sponsorship Program by CAST (2023QNRC001, S.G.); Textile Vision Basic Research Program of China (J202110, W.F.); Youth Innovation Team of Shaanxi Universities (W.F.); Institute of Flexible electronics and Intelligent Textile (W.F.); Ting Zhang and Xing Sun from Northwestern Polytechnical University for their guidance on numerical simulations.

## Author contributions

W.F. proposed and designed the whole process of the work. R.L. is responsible for the implementation of the experiment. H.D. contributed to the parameters of the electrospinning process. Z.W. and L.L. participated in and woven 3DPF. S.W. and X.L. supervised the piezoelectric experiment. W.C. participated in the discussion of the wear resistance test. M.R., S.G. and Y.L. contributed to MMT testing as well as the guidance of manuscripts. W.F. and R.L. wrote the manuscript. T.A. guided the revision of the paper. All authors provided insights into the analysis of experimental data and participated in the review of the manuscript.

## Ethics

All volunteers provided written informed consent.

## Competing interests

The authors declare no competing interests.
