## [Peer Review File · Nature Communications]

REVIEWER COMMENTS

Reviewer #1 (Remarks to the Author):

This paper designed and weaved a new kind of self-powered fabric 3DPF with the ultrahigh strength by hot stretching. The feature of anti-gravity unidirectional liquid transport was found due to the material and structural designs of fabrics. However, the manuscript cannot be accepted for publication in Nature Communications for the following reasons:

- 1.The solid evidence for the proposed mechanism of anti-gravity unidirectional liquid transport is not clear and further discussion needs to be provided.
- 2.The reasons corresponding to improving the piezoelectric property with sweat are not rational and scientific.
- 3.There exists various characteristics of this new fabric, but this work is not systematic and needs to pay more attention on the most important one. Besides, the application cannot exhibit the main advantages of the special fabric, including the ultrahigh strength and anti-gravity unidirectional liquid transport.

Here are the detailed comments:

- 1.In part 2.1, the authors mentioned that the hot stretching temperature should be chosen between the T_g and T_m of the polymer. But only 100 °C and 140 °C were explored. So why did you choose these two temperatures or you need to investigate more in order to ensure that 140 °C is the best stretching condition.
- 2.The authors have done lots of experiments to present the anti-gravity unidirectional liquid transport and only the possible generated forces and their directions are discussed in part 2.2. However, the internal mechanisms of creating forces should be illustrated in detail via conducting other experiments or designing mechanical model to provide more convincing evidence to verify your explanation. Besides, whether the phenomenon can be applied to the other structures consisting of hydrophilic outer layer and hydrophobic inner layer.
- 3.The authors illustrated that sweating promoted the electrical output of the 3DPF. However, the mechanism interpretation is not scientific. The electrical output of the piezoelectric device is originated from the induced charge rather than electron flow. The faster electron transport of piezoelectric layer would not lead to electrical performance enhancement. In addition, the sweat will be transported to the outer layer and not wet the PVDF layer. Therefore, I can hardly agree with your declaration.
- 4.In Figure 4e, the output waveform possesses multiple oscillating peaks, which is different from the normal piezoelectric signal. Whether the output comes entirely from the piezoelectricity of the material?
- 5.The sensitivity should be fitted through a series of data. Three points shown in Figure 4d are not enough to show the relationship between output performance and forces.

6.The author deduced that the sweat can reduce the resilience of the 3DPF, and lead to decreased output performances when force increasing. There is no basis for this inference. How about the output voltage of the 3DPF when lowering the patting frequency to ensure adequate recovery time?

7.The writing style of this article is not academic and lack of preciseness, for example, the sentence “the reasons ... may be related to ...” is not strict.

Reviewer #2 (Remarks to the Author):

This manuscript presents systematic research on the development of a PVDF piezoelectric nanoyarns strain sensor with ultrahigh strength. As a positive point, the reviewer would say that the bibliography is well done, the text is written in very clear and understandable English, even though we did not check the existing bibliography.

In a comprehensive review manuscript, it is expected to provide an insight for the existed articles in the field of this manuscript and provide authors input in experimental parts of the reviewed articles. How many samples were prepared for characterization and measurements? And if the numbers are stated, Is there any analysis which can prove the consistency of the achieved results? In these cases, a strong statistical analysis can be helpful. Therefor a close attention needs to perform on reviewing experimental research n this field by considering this point.

Also it is very important for some mentioned applications in the manuscript to have a uniform results over the several cycles. It would be great if authors could provide a section in this regards to review methods and approaches on fabrication of fibrous (or yarn) based sensors and nanogenerators with reliable uniformity of structure along the fiber?

In order to compare results, we have to know what the precision are and the accuracy of the measurements made for any characterization; no information is given according to these for results compared in this manuscript. Data processing reported for different published articles was not correct some times and the Figures are not well documented to provide a deep insight for the potential readers.

In the section of Piezoelectric performance enhancement of fabricated yarn, Where are the significant bands in FTIR spectra for all component and conformations for different piezoelectric materials? These includes the assignment of the bands for the three typical bands, particularly for the beta and gamma phase of PVDF fibers, for example. The problem is the same for all the figures and interpretations along the manuscript. I couldn't find out how multicomponent devices are really working as a part of the system. What is the role of each components These sections needs to be available in the revised version. So I strongly recommend they make serious modification for this part. Besides, authors may need also to cite relevant articles in this part to compare and justify their achieved results especially with those fabricated with PVDF polymers, (Some of them are listed below but not limited to them):

<https://pubs.acs.org/doi/10.1021/acsnm.0c01551>

<https://journals.sagepub.com/doi/pdf/10.1177/15280837211057575>

Enhanced the dielectric and piezoelectric properties of polyacrylonitrile piezoelectric composite fibers filled with ionic liquids - Shi - 2023 - Journal of Applied Polymer Science - Wiley Online Library

Maximizing Polyacrylonitrile Nanofiber Piezoelectric Properties through the Optimization of Electrospinning and Post-thermal Treatment Processes | ACS Applied Polymer Materials.

<https://journals.sagepub.com/doi/pdf/10.1177/1528083720928822>

<https://pubs.rsc.org/en/content/articlelanding/2021/xx/d2tc01931k/unauth>

<https://www.sciencedirect.com/science/article/abs/pii/B9780128206294000060>

<https://link.springer.com/article/10.1007/s10965-015-0765-8>

<https://link.springer.com/article/10.1007/s12274-021-3330-8>

<https://journals.sagepub.com/doi/abs/10.1177/1528083719867443>

<https://onlinelibrary.wiley.com/doi/full/10.1002/mame.202300009>

<https://pubs.rsc.org/en/content/articlehtml/2023/ra/d2ra06774a>

<https://onlinelibrary.wiley.com/doi/full/10.1002/mame.202200442>

What is the role of electrodes in a nanogenerator and electrical improvements? It is strongly recommended to provide a separate section for this part and evaluate the results:

<https://www.sciencedirect.com/science/article/abs/pii/S2589234722001257>

Enhancing the current density of a piezoelectric nanogenerator using a three-dimensional intercalation electrode | Nature Communications

Please provide some data for the conductive electrodes used for the different experiments available in the literature.

Once again, we know that the uniformity of the materials in the nanogenerators should affect the performance of nanogenerators. How many specimens were tested to be sure about the consistency and repeatability of the achieved results?

Altogether, I found this manuscript very informative and interesting for publication but revising aforementioned comments are required for final decision.

Reviewer #3 (Remarks to the Author):

The manuscript focuses on the development of a 3D piezoelectric fabric sensor weaved in a way to provide more comfort to users. The reported sensor showed the highest tensile strength among reported flexible piezoelectric sensors and has the capability to transport liquid from the inner layer to the outer layer. While there has been a lot of research on flexible piezoelectric sensors, the reviewer believes that the findings reported makes it appropriate for publication after major revision.

The authors use the hygroscopic properties in various layers for sweat permeability. The reviewer wonders what other originality is there to the research other than incorporating ultrahigh strength nanoyarn. The paper needs to highlight more of its originality in order for publication.

Page 7: What is the minimum requirement for 3D automatic looms? Did previous PVDF nanoyarns not have tensile strength to be applied to 3D automatic looms? Please elaborate on this part more to show the significance of the results.

Page 12: Please provide numerical difference in value or percentage difference between original 3DPF output signal and the dry 3DPF after sweat has evaporated.

Page 15: 3DPF sensor provides comfort to the user due to unidirectional water transport properties. However, most people who sweat after exercise would put their clothes in the laundry. Therefore, for durability testing, the reviewer wonders what happens to the 3DPF's efficiency after it goes through laundry. If the 3DPF fabric sensor is not capable of washing, the reviewer would like to know how it can be cleaned.

Page 15: The author also would like to know if after several cycles of sweating and drying, if there is odor that remains. If there isn't any odor, it could be mentioned in the section on comfort.

Page 16: While the 3DPF itself might look no different from typical fabric, the circuit connected to it could make it obvious that it can send emergency signals. The reviewer is wondering that realistically, would parents purchase these belts or rather buy them bracelets, necklaces, and toys as alternatives. The reviewer is not persuaded by this application as belts, which are usually made from leather, plastic, metal parts, etc., do not necessarily need special fabric to perform the emergency function.

Page 17: Several grammatical errors were spotted in the conclusion paragraph. The reviewer suggests the authors to read through the paper several times to remove these errors.

Reviewers' comments:

Reviewer #1

This paper designed and weaved a new kind of self-powered fabric 3DPF with the ultrahigh strength by hot stretching. The feature of anti-gravity unidirectional liquid transport was found due to the material and structural designs of fabrics. However, the manuscript cannot be accepted for publication in Nature Communications for the following reasons:

1. The solid evidence for the proposed mechanism of anti-gravity unidirectional liquid transport is not clear and further discussion needs to be provided.

Answer: Thank you for your comments. Based on your suggestion, the anti-gravity unidirectional liquid transport mechanism has been further clarified in the manuscript.

The 3DPF with a multi-layer structure can be divided into upper, middle, and lower layers according to the type of main yarn used (Fig. 1). The upper layer (blue yarns in Fig. 1) is composed of polyester fiber, the middle layer (gray yarns in Fig. 1) is mainly made of PVDF yarn while the lower layer (pink yarns in Fig. 1) is composed of viscose fiber.

Fig. 1. (a) Structure diagram, (b) cross-section, (c) top surface and (d) bottom surface of 3DPF, respectively.

The water absorption of viscose yarn is 708.2 %, which is more than twice that of polyester yarn (298.6%). While Coolmax yarn has a lower absorption rate than viscose, it still exceeds polyester fibers. On the other hand, PVDF yarn exhibits a much lower water absorption rate (69.4 %) than viscose and polyester yarn (Fig. 2). The Coolmax

yarns in 3DPF are oriented in a Z-direction and positioned perpendicular to the fabric surface. This unique configuration results in a strong core suction effect (Fig. 3) due to the multi-grooved surface structure of the Coolmax yarn (Fig. 4). It should be emphasized that water absorption of the fiber represents the maximum water content that the fiber can absorb, which reflects the hygroscopic properties of the fiber. Conversely, the core absorption height of the fiber is a parameter that characterizes the axial diffusion rate of water along the fiber.

Fig. 2. Comparison the water absorption of Coolmax yarn with polyester yarn, viscose yarn, conductive yarn and PVDF yarns.

Fig. 3. Comparison the wicking height of Coolmax yarn with polyester yarn, viscose yarn and PVDF yarns.

Fig. 4. Surface morphology and cross-sectional morphology of the Coolmax yarn.

The scale bar is 30 μm .

The findings reveal that polyester yarn and PVDF nanoyarn exhibit hydrophobic characteristics with water contact angles of 130.7° and 119° , respectively (Fig. 5a and Fig. 5b). Conversely, viscose yarn possesses hydrophilic properties with a water contact angle of 87.8° (Fig. 5c). Furthermore, a gradient wetting effect is observed in the thickness direction of the 3DPF owing to the increasing hydrophilic performance from the polyester layer to the PVDF layer followed by the adhesive layer.

Fig. 5. Water contact angle of (a) polyester yarn, (b) PVDF nanoyarn and (c) viscose yarn.

The unidirectional liquid transport mechanism of the 3DPF is shown in Fig. 6. When water is dropped on the polyester yarns side of the 3DPF, it follows the path of the Z-oriented yarns rather than absorbed by the polyester yarns and overcomes gravity. This is because the Z-oriented yarns made of Coolmax material possess greater core

absorption and water absorption capacity (force CF1) than polyester yarns. The strong capillary force of Z-oriented yarns propels the droplets upward. The water conveyed by the Z-oriented yarns from the bottom passes through the PVDF yarn layer, which has weak water absorption and is subsequently absorbed by the adhesive yarns on the upper layer of the 3DPF due to their super water absorption capacity (force CF2) (Fig. 6). Upon touching the upper layer, the droplet experiences an upward hydrophobic force (HF) imparted by the inner layer. Additionally, there is also a capillary force CF2 on the upper layer in the vertical and horizontal direction, which causes the droplet to spread laterally and subsequently wet the upper layer. Then, the droplet is transported to the upper layer without reverse transport, keeping the inner layer dry.

Fig. 6. Diagrams showing the simplified unidirectional water transport mechanism of the 3DPF.

In addition, the anti-gravity unidirectional liquid transport phenomenon of the 3DPF has been investigated numerically using the COMSOL Multiphysics simulation software. The level set method was adopted to capture the three-dimensional gas-liquid interface. The contact angle between water droplets and fabric fibers on the wetted wall is determined experimentally. Figures 7a and 7b show the water droplet transportation process in the 3DPF structure under the isometric and main view angles respectively. The complete videos are attached in Supplementary Movie 3 and Supplementary Movie 4. It can be seen that the simulation results are in agreement with the experimental

results. The droplet moves upward and contacts the polyester fiber first. This initial contact results in a decrease in droplet velocity due to contact resistance and gravity. However, upon the droplet meet the viscose fiber, a significant flow acceleration can be observed, which is caused by surface tension of the viscose fiber. The total simulation time is 0.01s, and the transport process can reach a stable state at around 0.008s. After that, the droplet is locked in the viscose yarn without change. It should be noted that the model and time scale are smaller, but the simulation process is the same as the actual physical process.

Fig. 7. The anti-gravity unidirectional liquid transport phenomenon in the 3DPF in (a) isometric view and (b) main view angles.

Fig. 8 illustrates a similar single-guide wet mechanism reported in the literature [Adv. Mater 2022, 34, 2106570]. It is observed that sweat generated on the lower side accumulates in the gap between the inner layer and the skin surfaces. The accumulated sweat is able to be rapidly pumped to the upper outer layer due to the pressure difference, with no reverse transport, ensuring that the inner layer in contact with the skin remains dry and comfortable at all times.

Fig. 8. Diagrams showing the simplified mechanism of the passive wicking process of poly (ionic liquid) (PIL) membrane (inset shows the process in the vertical direction).

We have added the corresponding figures and movies in the revised Supporting Information and modified the sentences in the revised manuscript (*line 2, page 10*).

(*line 2, page 10*) “The superior unidirectional liquid transportation ability of 3DPF is attributed to its inherent asymmetrical wettability and the strong core suction ability of the Z-oriented yarns with irregular section. The water absorption of viscose yarn is 708.2 %, which is more than twice that of polyester yarn (298.6%). While Coolmax yarn has a lower absorption rate than viscose, it still exceeds polyester fibers. On the other hand, PVDF yarn exhibits a much lower water absorption rate (69.4 %) than viscose and polyester yarn (Fig. S1). The Coolmax yarns in 3DPF are oriented in a Z-direction and positioned perpendicular to the fabric surface ³¹. This unique

configuration results in a strong core suction effect (Fig. S2) due to the multi-grooved surface structure of the Coolmax yarn (Fig. S3).

The findings reveal that polyester yarn and PVDF nanoyarn exhibit hydrophobic characteristics with water contact angles of 130.7° and 119° , respectively (Fig. S4a and Fig. S4b). Conversely, viscose yarn possesses hydrophilic properties with a WCA of 87.8° (Fig. S4c). Furthermore, a gradient wetting effect is observed in the thickness direction of the 3DPF owing to the increasing hydrophilic performance from the polyester layer to the PVDF layer, followed by the adhesive layer. The unidirectional liquid transport mechanism of the 3DPF is shown in Fig. 3h. When water is dropped on the polyester yarns side of the 3DPF, it follows the path of the Z-oriented yarns and overcomes gravity. This is because the Z-oriented yarns made of Coolmax material possess greater core absorption and water absorption capacity (force CF1) than polyester yarns. The strong capillary force of Z-oriented yarns propels the droplets upward. The water conveyed by the Z-oriented yarns from the bottom passes through the PVDF yarn layer, which has weak water absorption and is subsequently absorbed by the adhesive yarns on the upper layer of the 3DPF due to their super water absorption capacity (force CF2). Upon touching the upper layer, the droplet experiences an upward hydrophobic force (HF) imparted by the inner layer. Additionally, there is also a capillary force CF2 on the upper layer in the vertical and horizontal direction, which causes the droplet to spread laterally and subsequently wet the upper layer. The droplet is transported to the upper layer without reverse transport, keeping the inner layer dry. The anti-gravity unidirectional liquid transport phenomenon of the 3DPF has been investigated numerically using the COMSOL Multiphysics simulation software, and

the simulation results agree with the experimental results (Fig. S5, Supplementary Movie 3 and Supplementary Movie 4).”

2. *The reasons corresponding to improving the piezoelectric property with sweat are not rational and scientific.*

Answer: Thank you for your valuable comment. Based on the reviewer’s suggestion, we have repeated and verified the experiment results. We have also significantly improved the mechanism analysis to reinforce the accuracy of our findings.

All piezoelectric materials are dielectric and present an insulating state. However, a suitable moisture amount helps to increase the ferro-/piezoelectric performance of piezoelectric materials [*Sens. Actuators, A* 2010, **158**, 106; *Sens. Actuators, A* 2011, **167**, 19; *J. Biomechanics* 1976, **9**, 495; *Nature* 1970, **227**; *JCS-Japan* 1994, **102**, 537]. This is because the water molecular adsorption of the piezoelectric materials is a multiple physical and chemical process, which results in the transfer of concomitant electrons from the material [*Nano Lett.* 2009, **9**, 3720; *J. Chem. Phys. C* 2014, **118**, 15910; *J. Am. Chem. Soc.* 2007, **129**, 15684] [*J. Appl. Phys.* 2015, **118**, 15]. For example, the ferroelectric polarity and the piezoelectric performance of piezoelectric materials (e.g., LiNbO₃, BaTiO₃) were changed when it was exposed to liquid water, even sweat [*Nano Lett.* 2009, **9**, 3720; *Nano Lett.* 2016, **16**, 2400].

Another example is that the construction of chemical bonds occurs at the surface of BiFeO₃ in an aqueous solution, which can lead to a change in ferroelectric polarization. This is because sufficient polarization-selective chemical bonds at the ferroelectric

material surface were formed when exposed to the salt ion concentration in the sweat. The ionic introduction of salt ions on the surface can increase the electrostatic energy. At the BaTiO₃-water interface, the adsorption of cation ions from the salt liquid of the sweat along with chemical reactions, can lead to controllable chemical bonds. The large surface electric potential creates a strong atomic displacement of the Ti atom, causing it to move to the oxygen octahedral structure. As a result, metal electrode-O-H at the BTO piezoelectric sample surface is formed, leading to excellent piezoelectric properties of piezo-semiconductor materials [*Nature* 2014, **514**, 470; *Nat. Rev. Mater.* 2016, **1**, 7; *Nat. Commun.* 2021, **12**, 3508; *Nature* 2022, **608**, 69].

In order to ensure the validity and precision of the experiment, relevant tests have been conducted 10 times on five 3DPF samples with the same specifications. The results of the experiment confirmed that both the piezoelectric voltage and current output increased in the wet state. These findings are consistent with the relevant literature. The sweat is a kind of charge carrier. Xiong et al. proved that the piezoelectricity of the piezo-material significantly improved by introducing defect carriers [*J. Am. Chem. Soc.* 2023, **145**, 1936]. The insulated PVDF yarn can be considered a piezo-semiconductor material in suitable humidity conditions such as wetting by sweat. It should be noted that the PVDF yarn layer in this work only adsorbs a small amount of sweat due to its weak moisture absorption performance when the sweat is transferred from the Z-yarn to the viscose fiber at the top of 3DPF. Therefore, it can be concluded that the improvement of piezoelectric properties of 3DPF in the wet state is consistent with the existing literature.

This water-induced enhancement of ferroelectric polarization [*Nat. Commun* 2018, **9**, 3809] and piezoelectric performance [this work] opens up new opportunities for piezoelectric PVDF/BaTiO₃ textile composite in sensing applications.

In the revised manuscript, we have summarized the mechanism of the increase of piezoelectric properties caused by sweat.

3. There exists various characteristics of this new fabric, but this work is not systematic and needs to pay more attention on the most important one. Besides, the application cannot exhibit the main advantages of the special fabric, including the ultrahigh strength and anti-gravity unidirectional liquid transport.

Answer: The 3D textile structure piezoelectric sensor is designed with comfort, high sensitivity, mechanical durability and excellent unidirectional liquid transport performance for the human body (Fig. 6). The greatest significance of this paper is to propose a new strategy that utilizes fabric structure for sealing devices, which addresses the flexible electronic products are not breathable and can not be worn for a long time due to the traditional sealing process and makes a solid step forward for the practical application of flexible electronics. The highlights of the paper are as follows:

Fig. 6. Schematic diagram of 3DPF.

(1) The 3DPF has excellent anti-gravity unidirectional water transport. This is achieved through the hygroscopic effect of distinct yarn layers and the capillary force of Z yarn. The inner layer near the skin effectively transports sweat to the outer layer in just 4 s, ensuring a dry and comfortable experience.

(2) The tensile strength of the PVDF piezoelectric nanoyarns (313.3 MPa) obtained by conjugated electrostatic spinning and hot stretching method is better than those of the other PVDF and its copolymer nanoyarns. Meanwhile, the tensile strength of the 3DPF is the highest among all reported PVDF piezoelectric materials, reaching 46.0 MPa.

(3) Human sweating can positively impact the piezoelectric properties of the 3DPF, enhancing its sensing sensitivity from 0.41 V kPa^{-1} in dry conditions to 3.95 V kPa^{-1} when wet. As a result, the 3DPF can be effectively utilized as a self-powered switch that individuals can wear for extended periods.

(4) Continuous high-strength PVDF piezoelectric nanoyarns can be obtained by hot stretching electrospinning nanoyarns, while advanced 3D weaving technology has a high degree of automation, thereby allowing mass production of the 3DPF.

Fig. 5g compares sweat accumulation on the skin surface for the commercial piezoelectric sensor and the 3DPF after human sweating. The results reveal that significant sweat accumulates between the commercial PVDF piezoelectric sensor and the skin after exercising, while the skin beneath the 3DPF remains dry due to its unidirectional water transport. This finding highlights the commercial value of the 3DPF with anti-gravity transmission and its potential use as a component of a sports bracelet.

Fig. 5g. Comparison of the accumulation of sweat on the skin surface for commercial piezoelectric sensor and the 3DPF after human sweating.

In Fig. 6c, the 3DPF sheet that functions as a sickbed calling requires high strength and excellent durability. Similarly, the belt made of 3DPF (Fig. 6d) needs to have high strength and anti-gravity unidirectional liquid transport function to ensure the durability of the piezoelectric sensor.

Fig. 6. (c) The 3DPF is used as a sheet with the function of sickbed calling. (d) The 3DPF is used as a belt for missing children’s alarms and location.

3DPF exhibits outstanding unidirectional water transport and mechanical properties, thus being a promising candidate for use in bed sheets, watchbands, backpack belts, insoles, and other functional textiles for outdoor sports. Excessive human sweat can be absorbed, pumped, and released quickly from the skin through 3DPF, which would ensure the comfort of human wear.

Based on the reviewers' suggestions, we have enhanced the application description of 3DPF in the revised manuscript (*line 4, page 18*). Specifically, we have further emphasized its ultrahigh strength and anti-gravity unidirectional liquid transport capabilities. We believe these enhancements will help us better convey this application's unique and valuable features to our readers. Thank you for your feedback and suggestions, which have contributed to the overall quality of our work.

(line 4, page 18) “The 3DPF can be utilized as a bedsheet with a trigger switch conveniently integrated on its edge. This feature enables patients to call for assistance without getting up by pressing the local trigger switch embedded in the sheet. The ward call signal can be transmitted through Wi-Fi to remote medical staff, ensuring timely

care for patients. Additionally, the 3DPF sheet can achieve unidirectional fluid transport function, preventing bedsores, maintaining skin and sheet surface dryness and ensuring patient comfort, especially in cases of urinary incontinence (Fig. 6c).

The piezoelectric yarn placed inside the 3DPF is discreet and resembles commercial fabric, making it an ideal component for creating a hidden switch on a 3DPF belt of a child to signal distress in case of an abduction (Fig. 6d). Unlike other wearable accessories such as watch bands or necklaces, 3DPF is less conspicuous and not easily removable by potential abductors. 4G module (WH-GM5), GPS module (BD-126ZR) and others were used to develop the wireless alarm positioning system based on STM32 to realize the real-time transmission of signal and location information in the wide area network. The 4G circuit mainly packages the product status information and piezoelectric sensor signals for external communication. Notably, the 3DPF belt is comfortable and undetectable. When the child is lost or in danger, the child can trigger the alarm switch by pressing the belt. This will subsequently trigger a distress signal and real-time location information to be transmitted to the parent's device via a 4G network, enabling prompt intervention and rescue if needed (Supplementary Movie 8).”

4. In part 2.1, the authors mentioned that the hot stretching temperature should be chosen between the T_g and T_m of the polymer. But only 100 °C and 140 °C were explored. So why did you choose these two temperatures or you need to investigate more in order to ensure that 140 °C is the best stretching condition.

Answer: Thank you very much for your comments. The hot stretching temperature should be higher than the glass transition temperature of PVDF but lower than its melting temperature. The research was conducted on the as-spun PVDF nanoyarns, revealing that their T_m is approximately 165 °C (Fig. 2d). Initially, hot stretching temperatures of 100 °C and 140 °C were chosen, which increased the mechanical properties of the HS yarns. However, your suggestion prompted the exploration of HS yarns obtained under 80 °C and 150 °C (Fig. 2e), which also showed promising results. Before 140 °C, HS yarn mechanical properties positively correlate with increasing temperature. However, once the temperature rises to 150 °C, the tensile strength of HS yarn becomes lower than that at 140 °C. This phenomenon occurs due to the intensified thermal motion of the macromolecular chains in the fiber, leading to a relative slip and a subsequent reduction in breaking strength. Hence, it can be concluded that the optimal stretching condition for HS yarn is 140 °C.

We have added relevant sentences in the revised manuscript (*line 7, page 7*) and replaced the Fig. 2e.

Fig. 2d. Differential scanning calorimetry (DSC) data of PVDF as-spun nanoyarn.

Fig. 2e. Stress-strain curves of hot stretching nanoyarns (HS yarn) at different hot stretching conditions.

(line 7, page 7) “Before 140 °C, HS yarn mechanical properties positively correlate with increasing temperature. However, once the temperature surpasses 150 °C, the tensile strength of HS yarn becomes lower than that at 140 °C. This phenomenon occurs due to the intensified thermal motion of the macromolecular chains in the fiber, leading to a relative slip and a subsequent reduction in breaking strength. Hence, it can be concluded that the optimal stretching condition for HS yarn is 140 °C.”

5. *The authors have done lots of experiments to present the anti-gravity unidirectional liquid transport and only the possible generated forces and their directions are discussed in part 2.2. However, the internal mechanisms of creating forces should be illustrated in detail via conducting other experiments or designing mechanical model to provide more convincing evidence to verify your explanation. Besides, whether the phenomenon can be applied to the other structures consisting of hydrophilic outer layer and hydrophobic inner layer.*

Answer: Thank you very much for the suggestions.

The 3DPF with a multi-layer structure can be divided into upper, middle, and lower layers according to the type of main yarn used (Fig. 1). The upper layer (blue yarns in Fig. 1) is composed of polyester fiber, the middle layer (gray yarns in Fig. 1) is mainly made of PVDF yarn while the lower layer (pink yarns in Fig. 1) is composed of viscose fiber.

Fig. 1. (a) Structure diagram, (b) cross-section, (c) top surface and (d) bottom surface of 3DPF, respectively.

The water absorption of viscose yarn is 708.2 %, which is more than twice that of polyester yarn (298.6%). While Coolmax yarn has a lower absorption rate than viscose, it still exceeds polyester fibers. On the other hand, PVDF yarn exhibits a much lower water absorption rate (69.4 %) than viscose and polyester yarn (Fig. 2). The Coolmax yarns in 3DPF are oriented in a Z-direction and positioned perpendicular to the fabric surface. This unique configuration results in a strong core suction effect (Fig. 3) due to the multi-grooved surface structure of the Coolmax yarn (Fig. 4). It should be emphasized that water absorption of the fiber represents the maximum water content that the fiber can absorb, which reflects the hygroscopic properties of the fiber.

Conversely, the core absorption height of the fiber is a parameter that characterizes the axial diffusion rate of water along the fiber.

Fig. 2. Comparison the water absorption of Coolmax yarn with polyester yarn, viscose yarn, conductive yarn and PVDF yarns.

Fig. 3. Comparison the wicking height of Coolmax yarn with polyester yarn, viscose yarn and PVDF yarns.

Fig. 4. Surface morphology and cross-sectional morphology of the Coolmax yarn.

The scale bar is 30 μm .

The findings reveal that polyester yarn and PVDF nanoyarn exhibit hydrophobic characteristics with water contact angles of 130.7° and 119° , respectively (Fig. 5a and Fig. 5b). Conversely, viscose yarn possesses hydrophilic properties with a water contact angle of 87.8° (Fig. 5c). Furthermore, a gradient wetting effect is observed in the thickness direction of the 3DPF owing to the increasing hydrophilic performance from the polyester layer to the PVDF layer followed by the adhesive layer.

Fig. 5. Water contact angle of (a) polyester yarn, (b) PVDF nanoyarn and (c) viscose yarn.

The unidirectional liquid transport mechanism of the 3DPF is shown in Fig. 6. When water is dropped on the polyester yarns side of the 3DPF, it follows the path of the Z-oriented yarns rather than absorbed by the polyester yarns and overcomes gravity. This is because the Z-oriented yarns made of Coolmax material possess greater core absorption and water absorption capacity (force CF1) than polyester yarns. The strong capillary force of Z-oriented yarns propels the droplets upward. The water conveyed by the Z-oriented yarns from the bottom passes through the PVDF yarn layer, which has weak water absorption and is subsequently absorbed by the adhesive yarns on the upper layer of the 3DPF due to their super water absorption capacity (force CF2) (Fig. 6). Upon touching the upper layer, the droplet experiences an upward hydrophobic force

(HF) imparted by the inner layer. Additionally, there is also a capillary force CF2 on the upper layer in the vertical and horizontal direction, which causes the droplet to spread laterally and subsequently wet the upper layer. Then, the droplet is transported to the upper layer without reverse transport, keeping the inner layer dry.

Fig. 6. Diagrams showing the simplified unidirectional water transport mechanism of the 3DPF.

In addition, the anti-gravity unidirectional liquid transport phenomenon of the 3DPF has been investigated numerically using the COMSOL Multiphysics simulation software. The level set method was adopted to capture the three-dimensional gas-liquid interface. The contact angle between water droplets and fabric fibers on the wetted wall is determined experimentally. Figures 7a and 7b show the water droplet transportation process in the 3DPF structure under the isometric and main view angles respectively. The complete videos are attached in Supplementary Movies 3 and Supplementary Movie 4. It can be seen that the simulation results are in agreement with the experimental results. The droplet moves upward and contacts the polyester fiber first. This initial contact results in a decrease in droplet velocity due to contact resistance and gravity. However, upon the droplet meet the viscose fiber, a significant flow acceleration can be observed, which is caused by surface tension of the viscose fiber. The total simulation time is 0.01s, and the transport process can reach a stable state at around 0.008s. After that, the droplet is locked in the viscose yarn without change. It

should be noted that the model and time scale are smaller, but the simulation process is the same as the actual physical process.

Fig. 7. The anti-gravity unidirectional liquid transport phenomenon in the 3DPF in (a) isometric view and (b) main view angles.

Fig. 8 illustrates a similar single-guide wet mechanism reported in the literature [*Adv. Mater* 2022, 34, 2106570]. It is observed that sweat generated on the lower side accumulates in the gap between the inner layer and the skin surfaces. The accumulated sweat is able to be rapidly pumped to the upper outer layer due to the pressure difference,

with no reverse transport, ensuring that the inner layer in contact with the skin remains dry and comfortable at all times.

Fig. 8. Diagrams showing the simplified mechanism of the passive wicking process of poly(ionic liquid) (PIL) membrane (inset shows the process in the vertical direction).

The phenomenon in this work exhibits potential for application to other structures with a composition of hydrophilic outer layer and hydrophobic inner layer. Notably, the middle PVDF piezoelectric layer has been replaced with cotton yarn (R-3DPF), as depicted in Fig. 9. The resulting fabric also demonstrates the characteristic of unidirectional liquid transport.

It has been concluded that a three-way orthogonal structured fabric with a gradient wetting effect in the thickness direction results in a greater water absorption capacity of the top layer than the bottom layer. Moreover, the core suction effect of the Z-direction yarn is stronger than the bottom yarn, leading to a unidirectional liquid transport function with anti-gravity properties. However, it cannot be inferred that other structures exhibit similar properties without Z-yarn. Therefore, the innovative structural design of 3DPF is worthy of attention in this regard.

Fig. 9. (a) Directional water transport process from the top view of the R-3DPF. The water was penetrated from the inner layer to the viscose side. The tissue under the viscose layer was wetted. (b) Directional water transport process from the top view of the R-3DPF, the water was diffused onto the surface of the viscose side. The tissue under inner layer remained dry.

We have added the corresponding figures and movies in the revised Supporting Information and modified the sentences in the revised manuscript (*line 2, page 10*).

(*line 2, page 10*) “The superior unidirectional liquid transportation ability of 3DPF is attributed to its inherent asymmetrical wettability and the strong core suction ability of the Z-oriented yarns with irregular section. The water absorption of viscose yarn is 708.2 %, which is more than twice that of polyester yarn (298.6%). While Coolmax yarn has a lower absorption rate than viscose, it still exceeds polyester fibers. On the other hand, PVDF yarn exhibits a much lower water absorption rate (69.4 %) than viscose and polyester yarn (Fig. S1). The Coolmax yarns in 3DPF are oriented in a Z-direction and positioned perpendicular to the fabric surface ³¹. This unique

configuration results in a strong core suction effect (Fig. S2) due to the multi-grooved surface structure of the Coolmax yarn (Fig. S3).

The findings reveal that polyester yarn and PVDF nanoyarn exhibit hydrophobic characteristics with water contact angles of 130.7° and 119° , respectively (Fig. S4a and Fig. S4b). Conversely, viscose yarn possesses hydrophilic properties with a WCA of 87.8° (Fig. S4c). Furthermore, a gradient wetting effect is observed in the thickness direction of the 3DPF owing to the increasing hydrophilic performance from the polyester layer to the PVDF layer, followed by the adhesive layer. The unidirectional liquid transport mechanism of the 3DPF is shown in Fig. 3h. When water is dropped on the polyester yarns side of the 3DPF, it follows the path of the Z-oriented yarns and overcomes gravity. This is because the Z-oriented yarns made of Coolmax material possess greater core absorption and water absorption capacity (force CF1) than polyester yarns. The strong capillary force of Z-oriented yarns propels the droplets upward. The water conveyed by the Z-oriented yarns from the bottom passes through the PVDF yarn layer, which has weak water absorption and is subsequently absorbed by the adhesive yarns on the upper layer of the 3DPF due to their super water absorption capacity (force CF2). Upon touching the upper layer, the droplet experiences an upward hydrophobic force (HF) imparted by the inner layer. Additionally, there is also a capillary force CF2 on the upper layer in the vertical and horizontal direction, which causes the droplet to spread laterally and subsequently wet the upper layer. The droplet is transported to the upper layer without reverse transport, keeping the inner layer dry. The anti-gravity unidirectional liquid transport phenomenon of the 3DPF has been investigated numerically using the COMSOL Multiphysics simulation software, and

the simulation results agree with the experimental results (Fig. S5, Supplementary Movie 3 and Supplementary Movie 4).”

6. *The authors illustrated that sweating promoted the electrical output of the 3DPF. However, the mechanism interpretation is not scientific. The electrical output of the piezoelectric device is originated from the induced charge rather than electron flow. The faster electron transport of piezoelectric layer would not lead to electrical performance enhancement. In addition, the sweat will be transported to the outer layer and not wet the PVDF layer. Therefore, I can hardly agree with your declaration.*

Answer: Thank you for your valuable comment. We have repeated and verified the experiment results based on the reviewer's suggestion. We have also significantly improved the mechanism analysis to reinforce the accuracy of our findings.

All piezoelectric materials are dielectric and present an insulating state. However, a suitable moisture amount helps to increase the ferro-/piezoelectric performance of piezoelectric materials [*Sens. Actuators, A* 2010, **158**, 106; *Sens. Actuators, A* 2011, **167**, 19; *J. Biomechanics* 1976, **9**, 495; *Nature* 1970, **227**; *JCS-Japan* 1994, **102**, 537]. This is because the water molecular adsorption of the piezoelectric materials is a multiple physical and chemical process, which results in the transfer of concomitant electrons from the material [*Nano Lett.* 2009, **9**, 3720; *J. Chem. Phys. C* 2014, **118**, 15910; *J. Am. Chem. Soc.* 2007, **129**, 15684] [*J. Appl. Phys.* 2015, **118**, 15]. For example, the ferroelectric polarity and the piezoelectric performance of piezoelectric materials (e.g., LiNbO₃, BaTiO₃) were changed when it was exposed to liquid water, even sweat [*Nano Lett.* 2009, **9**, 3720; *Nano Lett.* 2016, **16**, 2400].

Another example is that the construction of chemical bonds occurs at the surface of BiFeO₃ in an aqueous solution, which can lead to a change in ferroelectric polarization. This is because sufficient polarization-selective chemical bonds at the ferroelectric material surface were formed when exposed to the salt ion concentration in the sweat. The ionic introduction of salt ions on the surface can increase the electrostatic energy. At the BaTiO₃-water interface, the adsorption of cation ions from the salt liquid of the sweat along with chemical reactions, can lead to controllable chemical bonds. The large surface electric potential creates a strong atomic displacement of the Ti atom, causing it to move to the oxygen octahedral structure. As a result, metal electrode-O-H at the BTO piezoelectric sample surface is formed, leading to excellent piezoelectric properties of piezo-semiconductor materials [*Nature* 2014, **514**, 470; *Nat. Rev. Mater.* 2016, **1**, 7; *Nat. Commun.* 2021, **12**, 3508; *Nature* 2022, **608**, 69].

In order to ensure the validity and precision of the experiment, relevant tests have been conducted 10 times on five 3DPF samples with the same specifications. The results of the experiment confirmed that both the piezoelectric voltage and current output increased in the wet state. These findings are consistent with the relevant literature. The sweat is a kind of charge carrier. Xiong et al. proved that the piezoelectricity of the piezo-material significantly improved by introducing defect carriers [*J. Am. Chem. Soc.* 2023, **145**, 1936]. The insulated PVDF yarn can be considered a piezo-semiconductor material in suitable humidity conditions such as wetting by sweat. It should be noted that the PVDF yarn layer in this work only adsorbs a small amount of sweat due to its weak moisture absorption performance when the sweat is transferred from the Z-yarn to the viscose fiber at the top of 3DPF. Therefore, it can be concluded that the

improvement of piezoelectric properties of 3DPF in the wet state is consistent with the existing literature.

This water-induced enhancement of ferroelectric polarization [*Nat. Commun* 2018, **9**, 3809] and piezoelectric performance [this work] opens up new opportunities for piezoelectric PVDF/BaTiO₃ textile composite in sensing applications.

In the revised manuscript, we have summarized the mechanism of the increase of piezoelectric properties caused by sweat.

7. In Figure 4e, the output waveform possesses multiple oscillating peaks, which is different from the normal piezoelectric signal. Whether the output comes entirely from the piezoelectricity of the material?

Answer: The output waveform possesses multiple oscillating peaks due to the AC excitation signal. In general, there is a delay between the piezoelectric output signal and the AC excitation signal. In the context of PVDF piezoelectric energy harvesting, it has been observed in the literature that the output peak also exhibits slight fluctuations [*Nanoscale* 2018, **10**, 17751; *Nano Energy* 2022, **99**, 107400; *Adv. Mater.* 2019, **29**, 1904020; *Adv. Mater.* 2017, **29**, 1702308], which corroborates our experimental findings, as demonstrated in Fig. 4e.

Piezoelectric materials are a type of dielectric material and insulator that possess a unique property of increased conductivity when exposed to AC excitation signals. This characteristic allows the piezoelectric materials to function as capacitors, blocking the

flow of DC current while allowing AC current to pass through. Therefore, 3DPF will generate current and voltage output under the pulsed AC excitation signal, hence, it has a high power output ($P = UI$) [Adv. Mater. 2019, 29, 1904020; Adv. Mater. 2017, 29, 1702308]. Accordingly, the output in this work comes entirely from the piezoelectricity of the material.

8. *The sensitivity should be fitted through a series of data. Three points shown in Figure 4d are not enough to show the relationship between output performance and forces.*

Answer: Thank you for your suggestion. The number of experiments has been increased and the relevant data has been added to Fig. 4d in the revised manuscript. Additionally, the sensitivity of the straight-line segment data has been analysed, revealing that the sensitivity of 3DPF is 0.41 V kPa^{-1} (under the low-pressure range).

Fig.4d. Voltage sensitivity of the 3DPF

9. *The author deduced that the sweat can reduce the resilience of the 3DPF, and lead to decreased output performances when force increasing. There is no basis for this*

inference. How about the output voltage of the 3DPF when lowering the patting frequency to ensure adequate recovery time?

Answer: Thank you for your comments. According to your suggestion, we have supplemented the output voltage of the 3DPF in a wet state at a low frequency. The output voltage of the 3DPF increases first and then decreases when the force increases (Fig. 10).

Fig.10. Piezoelectric outputs of 3DPF at 20 μ L sweat volume and excitation frequency of 1.8 Hz.

It is well-known that the piezoelectric properties of a material are impacted by its elastic properties. The fibers employed in this study are all polymer fibers. When they absorb water, they expand in volume. This expansion manifests in transverse expansion and longitudinal reduction. The underlying cause is the penetration of water molecules into the amorphous region of the fiber, which compresses the molecular spacing and induces bending in the molecular chain, thereby thickening the fiber and slightly elongating it longitudinally. However, the longitudinal expansion is not significant, resulting in a small change in length. In the thickness direction of 3DPF, the fabric contains yarn

bindings, which prevents an increase in overall thickness. As a result, fiber expansion reduces the gap between the fabric yarns and subsequently decreases the compressible space. Water in fibers leads to a decrease in strength, modulus, elasticity, and stiffness, ultimately resulting in poor fabric resilience. This is due to the hygroscopicity of the fiber, which weakens the interaction between the macromolecular chains, making them susceptible to conformational changes and slippage. The mechanical properties of the fiber deteriorate, and the application of high pressure can lead to intermolecular fracture and structural damage, which cannot be restored to their original state. Hence, the failure of fabric rebound cannot be attributed to a temporary cause but rather the excessive force that leads to the disruption of molecular chains, rendering the fabric irrecoverable regardless of the duration.

10. *The writing style of this article is not academic and lack of preciseness, for example, the sentence “the reasons ... may be related to ...” is not strict.*

Answer: Thank you for your suggestion. We have conducted a thorough experimental and theoretical analysis to enhance the scientific rigour of the article. The entire manuscript has been diligently reviewed and refined. We have enlisted the services of a professional language editor to ensure that the revised version is devoid of grammatical errors. In order to demonstrate our commitment to accuracy and quality, we are pleased to provide a proofreading certificate for your reference. The polish certificate is as follows:

Reviewer #2:

This manuscript presents systematic research on the development of a PVDF piezoelectric nanoyarns strain sensor with ultrahigh strength. As a positive point, the reviewer would say that the bibliography is well done, the text is written in very clear and understandable English, even though we did not check the existing bibliography.

Answer: Thank you very much for your compliments. We are very grateful for your positive comments on our work. The modified contents in the text are marked in red. The specific responses to comments are as follows:

1. *In a comprehensive review manuscript, it is expected to provide an insight for the existed articles in the field of this manuscript and provide authors input in experimental parts of the reviewed articles.*

Answer: Thank you very much for your comments. We would like to express our sincere gratitude for your insightful feedback. We have considered your comments carefully and incorporated the relevant literature in the appropriate sections of the revised manuscript.

2. *How many samples were prepared for characterization and measurements? And if the numbers are stated, is there any analysis which can prove the consistency of the achieved results? In these cases, a strong statistical analysis can be helpful. Therefore a close attention needs to perform on reviewing experimental research in this field by considering this point. Also it is very important for some mentioned applications in the manuscript to have an uniform results over the several cycles.*

Answer: Thank you for your comments. There are at least five samples prepared for characterization and measurements.

The consistency of the experimental data has been verified through ANOVA to provide further evidence of the consistency of the results. Error bars have also been added to the graph.

The wet and piezoelectric properties of plate 3DPF were re-tested after three months. No significant difference was observed between the experimental results and the original samples.

The above discussion has been added to the revised manuscript.

3. It would be great if authors could provide a section in this regard to review methods and approaches on fabrication of fibrous (or yarn) based sensors and nanogenerators with reliable uniformity of structure along the fiber?

Answer: Thank you for your valuable suggestions. A new section has been added to review methods and approaches to fabricating fibrous (or yarn) based sensors and nanogenerators with reliable uniformity of structure along the fiber (section 1).

4. In order to compare results, we have to know what the precision are and the accuracy of the measurements made for any characterization; no information is given according to these for results compared in this manuscript. Data processing reported for different

published articles was not correct some times and the Figures are not well documented to provide a deep insight for the potential readers.

Answer: Thank you for your comments. The precision of the measurements made for all characterization has been included in the revised manuscript (*line 14, page 20 and line 25, page 20*). The consistency of the experimental data has been verified through ANOVA to provide further evidence of the consistency of the results. Error bars have also been added to the graph.

(line 14, page 20) “According to the ISO13934-2: 1999 standard, the tensile strength was assessed by the universal testing machine (UTM5205X, Shenzhen, China). The accuracy level of the machine is 0.5. The force control rate accuracy and deformation control rate accuracy are both within $\pm 1\%$ of the set value when the rate is less than $0.05\% \text{ FS s}^{-1}$. When the rate is greater than $0.05\% \text{ FS s}^{-1}$, both are within $\pm 0.5\%$ of the set value. Additionally, the precision of the displacement control rate is within $\pm 0.5\%$ of the set value. The WCA was tested by a single-fiber contact angle meter (OCA40MICRO, Germany) and the accuracy of the video system is $\pm 0.1^\circ$.”

(line 25, page 20) “The piezoelectric data was recorded by an electrometer (Keithley 6514, USA), which reads 1200 readings per second. The 3DPF was washed in a rigorous laundering environment according to AATCC standard 135. A sample with the size of 20 cm^2 was used for the gas permeability test (Automatic permeability meter, YG461E-III, China), the measurement accuracy is 1 mm s^{-1} .”

5. In the section of Piezoelectric performance enhancement of fabricated yarn, where are the significant bands in FTIR spectra for all component and conformations for different piezoelectric materials? These includes the assignment of the bands for the three typical bands, particularly for the beta and gamma phase of PVDF fibers, for example. The problem is the same for all the figures and interpretations along the manuscript.

Answer: Thank you very much for the suggestions. The description of the FTIR spectra has been adjusted in the main content of the revised manuscript (**line 27, page 8**). PVDF is a linear semicrystalline polymer possessing five different crystalline phases, i.e., α , β , γ , δ , and ϵ . The α and δ phases show trans-gauche-trans-gauche (TG^+TG^-) chain conformations, β -phase has all trans conformation (TTTT) while γ and ϵ phases have trans-trans-trans-gauche-trans-trans-trans-gauche ($T_3G^+T_3G^-$) chain conformations. Among all these phases, the β -phase has the highest spontaneous polarization and, thereby, better piezoelectric properties than other phases. In contrast, the α phase is the most stable since it has the lowest energy. The piezoelectricity of PVDF is dependent upon its β -phase content as well as crystallinity. The higher the β -phase content and its crystallinity, the higher the piezoelectric coefficient [*Macromol. Mater. Eng.* 2022, **308**, 2200442]. Therefore, we mainly studied the changes in α and β phases of PVDF before and after hot stretching.

(line 27, page 8) “PVDF powder has five crystal forms: α , β , γ , δ , and ϵ phase, among which the most common crystal type is the nonpolar α phase⁴⁴. The α -crystal chain dipole is the opposite and does not show polarity, but it can be transformed into other phases, such as β phase under sufficient mechanical stress, heat or electricity^{45,46}. The

β -crystal cell contains polar zig-zag chains, which is the key to the piezoelectric properties of PVDF. The characteristic peaks at 764 and 975 cm^{-1} wavenumbers are α -phase, while those at 841 and 1275 cm^{-1} wavenumbers are β -phase⁴⁷. In this work, most of the α phase was polarized to β phase in the process of high voltage electrostatic spinning and hot stretching process (Fig. 2k). Therefore, characteristic peaks at 764 and 975 cm^{-1} of PVDF and HS yarn decrease, while those at 841 and 1275 cm^{-1} increase.”

Fig. 2. Morphology and properties of PVDF nanoyarns before and after hot stretching. a Stress-strain curves of PVDF as-spun nanoyarn at different collection speeds. **b-c** SEM image of PVDF as-spun nanoyarn. **d** Differential scanning calorimetry (DSC) data of PVDF as-spun nanoyarn. **e** Stress-strain curves of hot stretching nanoyarns (HS yarn) at different hot stretching conditions. **f** Photograph and **g-h** SEM images of HS yarn. **i** Maximum strength of PVDF nanoyarn and HS yarn. **j** Tensile strength comparison between HS yarn and the other PVDF and its copolymers nanoyarns. **k** FTIR and **l** XRD patterns of the HS yarn, PVDF nanoyarn and PVDF powder, respectively.

6. I couldn't find out how multicomponent devices are really working as a part of the system. What is the role of each components These sections need to be available in the revised version. So I strongly recommend they make serious modification for this part. Besides, authors may need also to cite relevant articles in this part to compare and justify their achieved results especially with those fabricated with PVDF polymers, (Some of them are listed below but not limited to them):

<https://pubs.acs.org/doi/10.1021/acsanm.0c01551>

<https://journals.sagepub.com/doi/pdf/10.1177/15280837211057575>

Enhanced the dielectric and piezoelectric properties of polyacrylonitrile piezoelectric composite fibers filled with ionic liquids - Shi - 2023 - Journal of Applied Polymer Science - Wiley Online Library
Maximizing Polyacrylonitrile Nanofiber Piezoelectric Properties through the Optimization of Electrospinning and Post-thermal Treatment Processes | ACS Applied Polymer Materials.

<https://journals.sagepub.com/doi/pdf/10.1177/1528083720928822>

<https://pubs.rsc.org/en/content/articlelanding/2021/xx/d2tc01931k/unauth>

<https://www.sciencedirect.com/science/article/abs/pii/B9780128206294000060>

<https://link.springer.com/article/10.1007/s10965-015-0765-8>

<https://link.springer.com/article/10.1007/s12274-021-3330-8>

<https://journals.sagepub.com/doi/abs/10.1177/1528083719867443>

<https://onlinelibrary.wiley.com/doi/full/10.1002/mame.202300009>

<https://pubs.rsc.org/en/content/articlehtml/2023/ra/d2ra06774a>

<https://onlinelibrary.wiley.com/doi/full/10.1002/mame.202200442>

Answer: The 3DPF devices are shown in Fig. 1 in the original manuscript. The structure of the 3DPF devices has been redrawn to demonstrate the components and their respective roles.

Fig. 1. a) Structure diagram, b) cross-section, c) top surface and d) bottom surface of 3DPF, respectively.

The PVDF nanoyarn possesses piezoelectric properties, which enable it to generate an electrical signal under pressure. The silver-nylon yarn is responsible for transmitting the electrical signals produced by PVDF fibers. Coolmax yarn, which has a profiled cross-section and excellent moisture conductivity, served as the medium for water transfer in this work. Viscose yarn composed of cellulose fibers with strong hygroscopic properties was used to absorb water in this study. Polyester yarn with high strength and water absorption capabilities was employed to provide strength and moisture transfer. The piezoelectric mechanism of the 3DPF is illustrated in Fig. 4a. The electrical signals generated by the PVDF piezoelectric layer are conveyed through the conducting yarns located above and below the piezoelectric layer in the 3DPF. We have added the information and the relevant literature in the revised manuscript (*line 26, page 5*).

(line 26, page 5) “The PVDF nanoyarn possesses piezoelectric properties, which enable it to generate an electrical signal under pressure. The silver-nylon yarn is responsible for transmitting the electrical signals produced by PVDF fibers. Coolmax yarn, which has a profiled cross-section and excellent moisture conductivity, served as the medium for water transfer in this work. Viscose yarn composed of cellulose fibers with strong hygroscopic properties was used to absorb water in this study. Polyester yarn with high strength and water absorption capabilities was employed to provide strength and moisture transfer.”

Fig. 4a. The piezoelectric mechanism diagram of the 3DPF.

The comparison of the performance of this work with other fiber-based piezoelectric sensors was described in line 16, page 15. The description in the original may not have been direct or prominent enough to be noticed, which could have led readers to ignore important content. In the revised manuscript, we have strengthened the description of this part (line 19, page 15) and provided detailed comparative data in Table S3 (Supporting information, page 2). The comparison reveals that the β -phase content and sensitivity of the 3DPF surpass that of many other fiber-based piezoelectric sensors.

(*line 19, page 15*) “Owing to the special structure of 3DPF, the sensitivity of 3DPF is higher than most of other PVDF-based piezoelectric sensors ^[1-6] (Table S3, Supporting information)”.

Table S3. Comprehensive property comparison of different flexible piezoelectric sensors.

Sample	Structure	β -phase content (%)	Sensitivity	Tensile strength (MPa)	Ref.
PVDF	3Dorganic woven fabric	88.08	0.41 V kPa ⁻¹ 3.95 V kPa ⁻¹ (1.025 V N ⁻¹ – 9.875 V N ⁻¹)	46.0 ± 4.3	This work
PVDF/MOF	Nanofibrous membrane	75	0.118 V N ⁻¹	< 25	[65]
(PVDF-BaTiO ₃)/PA-11	Nanofiber membrane	69.2	107.52 mV N ⁻¹	16.89±1.52 N	[66]
PVDF	Nanofiber webs	80.52	/	/	[67]
PVDF-ZnO	Nanofiber mats	87	/	/	[68]
PVDF/ZnO	Nanofiber mats	87	2.1795 mV N ⁻¹	/	[69]
PVDF/ZnO	Nanofiber mats	81.4	2.37 mV kPa ⁻¹	/	[70]

7. What is the role of electrodes in a nanogenerator and electrical improvements? It is strongly recommended to provide a separate section for this part and evaluate the results:

<https://www.sciencedirect.com/science/article/abs/pii/S2589234722001257>

Enhancing the current density of a piezoelectric nanogenerator using a three-dimensional intercalation electrode | *Nature Communications*

Please provide some data for the conductive electrodes used for the different experiments available in the literature.

Answer: Thank you for your suggestion. After thoroughly reviewing the relevant literature, we have included a new section that provides an overview of the impact of electrodes on piezoelectric output. Given that the intended application involves fabric weaving, we have opted to use silver nylon electrodes with exceptional electrical conductivity and softness. The new section has been incorporated into Section 1 (*line 16, page 5*).

(line 16, page 5) “Moreover, in the structure of the piezoelectric nanogenerator, the electrode is responsible for transferring the electrical energy generated by the piezoelectric material to other areas, such as load or electric consumers. Therefore, the selection, design, and manufacture of electrode materials are of utmost importance³². The commonly used electrodes include metal-based electrodes³³, carbon-based, ink-based, conductive polymer-based and conductive threads. Since the electrode in this work needs to be woven in fabric, silver-plated nylon yarn is an optimal selection due to its exceptional electrical conductivity, high strength, and remarkable flexibility³⁴”

8. *Once again, we know that the uniformity of the materials in the nanogenerators should affect the performance of nanogenerators. How many specimens were tested to be sure about the consistency and repeatability of the achieved results? Altogether, I found this manuscript very informative and interesting for publication but revising aforementioned comments are required for final decision.*

Answer: Thank you for your insightful and positive comments. According to the standard protocol for repeated testing, at least five specimens were tested to ensure the consistency and repeatability of the results obtained. The relevant sentences have been added to the revised manuscript (*line 4, Page 22*).

(line 4, Page 22) “According to the standard protocol for repeated testing, at least five specimens were tested to ensure the consistency and repeatability of the results obtained.”

Reviewer #3:

The manuscript focuses on the development of a 3D piezoelectric fabric sensor weaved in a way to provide more comfort to users. The reported sensor showed the highest tensile strength among reported flexible piezoelectric sensors and has the capability to transport liquid from the inner layer to the outer layer. While there has been a lot of research on flexible piezoelectric sensors, the reviewer believes that the findings reported makes it appropriate for publication after major revision.

Answer: We sincerely thank the reviewer for recognizing and acknowledging our work. We highly appreciate your valuable suggestions, which will certainly assist us in improving the quality of our paper. We have carefully reviewed and considered all of your comments and suggestions and provided detailed responses to each of them below.

1. The authors use the hygroscopic properties in various layers for sweat permeability. The reviewer wonders what other originality is there to the research other than incorporating ultrahigh strength nanoyarn. The paper needs to highlight more of its originality in order for publication.

Answer: Thank you very much for your comments. The 3D textile structure piezoelectric sensor is designed with comfort, high sensitivity, mechanical durability and excellent unidirectional liquid transport performance for the human body (Fig. 1). The highlights of the paper are as follows:

Fig. 1. Schematic diagram of 3DPF.

(1) The 3DPF has excellent anti-gravity unidirectional water transport. This is achieved through the hygroscopic effect of distinct yarn layers and the capillary force of Z yarn. The inner layer near the skin effectively transports sweat to the outer layer in just 4 s, ensuring a dry and comfortable experience.

(2) The tensile strength of the PVDF piezoelectric nanoyarns (313.3 MPa) obtained by conjugated electrostatic spinning and hot stretching method is better than those of the other PVDF and its copolymer nanoyarns. Meanwhile, the tensile strength of the 3DPF is the highest among all reported PVDF piezoelectric materials, reaching 46.0 MPa.

(3) Human sweating can positively impact the piezoelectric properties of the 3DPF, enhancing its sensing sensitivity from 0.41 V kPa^{-1} in dry conditions to 3.95 V kPa^{-1} when wet. As a result, the 3DPF can be effectively utilized as a self-powered switch that individuals can wear for extended periods.

(4) Continuous high-strength PVDF piezoelectric nanoyarns can be obtained by hot stretching electrospinning nanoyarns, while advanced 3D weaving technology has a high degree of automation, thereby allowing mass production of the 3DPF.

The design idea of 3DPF provides a new strategy for developing practical smart wearable products that are both comfortable and practical.

The description of originality in the abstract and conclusion of the revised manuscript (*line 7, Page 20*) has been improved.

(line 7, Page 20) “Furthermore, it is noteworthy that the piezoelectric properties of the 3DPF are actually enhanced by human sweating rather than weakened. Specifically, the 3DPF sensing sensitivity increases from 0.41 V kPa^{-1} in the dry state to 3.95 V kPa^{-1} in the wet state. In addition, the response time decreases from 100 ms in the dry state to 50 ms in the wet state. These remarkable properties render the 3DPF a potential self-powered switch, which can be worn by individuals for extended periods, triggering alarm signals or location information and transmitting them via Wi-Fi or a 4G module developed based on STM32, thus facilitating timely rescue. The proposed intelligent wearable fabric strikes a balance between comfort and sensing properties, making it feasible for intelligent wearable products to be worn by the human body for prolonged periods.”

2. Page 7: *What is the minimum requirement for 3D automatic looms? Did previous PVDF nanoyarns not have tensile strength to be applied to 3D automatic looms? Please elaborate on this part more to show the significance of the results.*

Answer: Thank you for your suggestion. It is worth noting that there is no established industry standard for minimum 3D automatic loom requirements. However, it is imperative to adhere to the rapier loom digital control system standard (FZ/T 99022-2023), which specifies that the weaving tension of a single yarn should not exceed 25% of its breaking strength. Our research has shown that PVDF nanoyarns without hot stretching are unsuitable for 3D automatic looms. This is because the fibers on the surface of the PVDF nanoyarns (Fig. 2b) is less axial orientation, and the hairiness is serious after repeated friction with the steel buckle during the weaving process. This hairiness leads to entanglement between yarns which hinders continuous weaving. On the other hand, HS yarns obtained after hot stretching have mechanical properties 900% higher than the original nanoyarns. Furthermore, most fibers on the surface are axially oriented (Fig. 2g), making them suitable for high-speed weaving on 3D automatic looms.

In addition, we have opted for high-performance HS yarns for the weaving process due to their enhanced piezoelectric properties.

We have added a description of the advantages of stretched yarns to the revised manuscript (*line 19, Page 8*).

Fig. 2. Morphology and properties of PVDF nanoyarns before and after hot stretching. **a** Stress-strain curves of PVDF as-spun nanoyarn at different collection speeds. **b-c** SEM image of PVDF as-spun nanoyarn. **d** Differential scanning calorimetry (DSC) data of PVDF as-spun nanoyarn. **e** Stress-strain curves of hot stretching nanoyarns (HS yarn) at different hot stretching conditions. **f** Photograph and **g-h** SEM images of HS yarn. **i** Maximum strength of PVDF nanoyarn and HS yarn. **j** Tensile strength comparison between HS yarn and the other PVDF and its copolymers nanoyarns. **k** FTIR and **l** XRD patterns of the HS yarn, PVDF nanoyarn and PVDF powder, respectively.

(line 19, Page 8) “Our research has shown that PVDF nanoyarns without hot stretching are unsuitable for 3D automatic looms. This is because the fibers on the surface of the PVDF nanoyarns (Fig. 2b) is less axial orientation, and the hairiness is serious after repeated friction with the steel buckle during the weaving process. This hairiness leads to entanglement between yarns which hinders continuous weaving.”

3. Page 12: Please provide numerical difference in value or percentage difference between original 3DPF output signal and the dry 3DPF after sweat has evaporated.

Answer: Thank you for your suggestion. We have provided the numerical difference in output value between the original 3DPF and the 3DPF after the sweat has evaporated in the revised manuscript (*line 17, page 14*).

(line 17, page 14) “When the sweat is evaporated, the output signal of the dry 3DPF is almost the same as the original 3DPF, and the peak volatility is less than 3% (Fig. S7).”

4. Page 15: 3DPF sensor provides comfort to the user due to unidirectional water transport properties. However, most people who sweat after exercise would put their clothes in the laundry. Therefore, for durability testing, the reviewer wonders what happens to the 3DPF’s efficiency after it goes through laundry. If the 3DPF fabric sensor is not capable of washing, the reviewer would like to know how it can be cleaned.

Answer: Thank you for your suggestions. We have tested the robustness and stability of the 3DPF by washing experiment according to AATCC standard 135, which is widely used to test the washable property of yarns or fabrics. Based on the results, it can be stated that the 3DPF exhibits excellent resistance to washing, with no discernible changes observed in its electrical performance after undergoing 10 rounds of washing experiments. The new descriptions have been added in section 2.4 (*line 5, page 16*) and section 4 (*line 26, page 21*), while the data is presented in Fig. S6.

(line 5, page 16) “...excellent mechanical durability of the 3DPF (Fig. 5d). Moreover, the property of being washable is a fundamental requirement for fabrics. The output voltage of 3DPF was tested after various washing cycles. As shown in Fig. S10, the output voltage of 3DPF remained consistent after ten washing cycles. The results suggest that 3DPF possesses an excellent washability.”

(line 26, page 21) “The 3DPF was washed in a rigorous laundering environment according to AATCC standard 135.”

Fig. S10. The electrical output of 3DPF after different washing times.

5. Page 15: The author also would like to know if after several cycles of sweating and drying, if there is odor that remains. If there isn't any odor, it could be mentioned in the section on comfort.

Answer: Thank you for your great suggestions. After several sweating and drying cycles, no odor was detected in the experiment. Human sweat comprises various organic metabolites, including glucose, lactic acid, cortisol, and uric acid [*Int. J. Electrochem. Sci.* 2022, **17**, 220534]. The odor is typically caused by the degradation

of organic substances during the piezoelectric process [*Nature* 1970, 227]. We are grateful for your valuable input and have incorporated the advantages you highlighted in our revised manuscript (*line 12, Page 17*). Thank you for your invaluable suggestions.

(line 12, Page 17) “After several sweating and drying cycles, no odor was detected in the experiment. Human sweat comprises various organic metabolites, including glucose, lactic acid, cortisol, and uric acid ⁷⁶. The odor is typically caused by the degradation of organic substances during the piezoelectric process ⁷⁷.”

6. Page 16: While the 3DPF itself might look no different from typical fabric, the circuit connected to it could make it obvious that it can send emergency signals. The reviewer is wondering that realistically, would parents purchase these belts or rather buy them bracelets, necklaces, and toys as alternatives. The reviewer is not persuaded by this application as belts, which are usually made from leather, plastic, metal parts, etc., do not necessarily need special fabric to perform the emergency function.

Answer: Thank you for your insightful comments. Belts designed for adults typically comprise leather, plastic, and metallic materials. Conversely, younger individuals tend to opt for fabric belts (Fig. 3), owing to their superior breathability and comfort compared to their leather counterparts. Children's belts are predominantly made of fabric (Fig. 4), due to their ease and comfort of wear.

The concealed alarm device implemented in the 3DPF belt is designed to evade detection by potential assailants. Initially, the plan was to transform the 3DPF belt into a wristwatch band (Fig. 5) for children. However, with the prevalence of smartphones

and watches, abductors or kidnappers can easily identify and dispose of conspicuous electronics. In contrast, the 3DPF belt made entirely of yarn is not easily detectable. Notably, few parents give necklaces or bracelets to children below ten, especially metal ones, and it is difficult to conceal the alarm device in them. Toys are also unreliable, as children can easily lose or discard them. Therefore, we consider the 3DPF belt a promising option for a belt with concealed alarm functionality, owing to its remarkable mechanical properties, piezoelectric transmission properties, and comfort.

Thanks for your suggestions. This 3DPF fabric with both a single-guide wet and high strength can be used in many applications, such as watch straps, backpack straps, and insoles (Fig. 6). We have added the relevant descriptions in the revised manuscript (*line 1, Page 19*).

(line 5, Page 18) “The piezoelectric yarn placed inside the 3DPF is discreet and resembles commercial fabric, making it an ideal component for creating a hidden switch on a 3DPF belt of a child to signal distress in case of an abduction (Fig. 6d). Unlike other wearable accessories such as watch bands or necklaces, 3DPF is less conspicuous and not easily removable by potential abductors. Unlike other wearable accessories such as watch bands or necklaces, 3DPF is less conspicuous and not easily removable by potential abductors. 4G module (WH-GM5), GPS module (BD-126ZR) and other modules were used to develop the wireless alarm positioning system based on STM32 to realize the real-time transmission of signal and location information in the wide area network. The 4G circuit mainly packages the product status information and piezoelectric sensor signals for external communication. Notably, the 3DPF belt is comfortable and undetectable. When the child is lost or in danger, the child can trigger

the alarm switch by pressing the belt. This will subsequently trigger a distress signal and real-time location information to be transmitted to the parent's device via a 4G network, enabling prompt intervention and rescue if needed (Supplementary Movie 8).”

Fig. 3. Adult belt

https://detail.tmall.com/item.htm?ali_refid=a3_430673_1006:1124498165:N:p7pMvdLEL13+v4jFP9MkCA==:af4662b4f35218043772fdc958ff15e8&ali_trackid=162_af4662b4f35218043772fdc958ff15e8&id=677701584215&spm=a2e0b.20350158.31919782.7

Fig. 4. Children's belt

<https://item.jd.com/10056803074564.html#none>

Fig. 5. Smartwatch with fabric belt

Fig. 6. The 3DPF used as insole, backpack, watchband

7. Page 17: Several grammatical errors were spotted in the conclusion paragraph.

The reviewer suggests the authors to read through the paper several times to remove these errors.

Answer: Thank you for your suggestions. We have undertaken a thorough examination of the entire document and engaged a professional language editor to ensure that the revised manuscript is devoid of any grammatical errors. We are pleased to share the proofreading certificate as evidence of this commitment to quality assurance.

The polish certificate is as follows:

REVIEWER COMMENTS

Reviewer #2 (Remarks to the Author):

I write you in regards to the revised version of the Manuscript ID: NCOMMS-23-26669 entitled "Sweat permeable and ultrahigh strength 3D PVDF piezoelectric nanoyarn fabric strain sensor".

Authors have done an interesting work to develop a fundamental aspects of fabrication of textile based piezoelectric materials and providing a systematic research on different approaches to maximize the piezoelectric performance of the developed yarn structures using the conjugate electrospinning following by the post treatments. Authors have also mentioned different set up to improve the piezoelectric properties of PVDF based fibers. Moreover, they succeed to provide an image to their potential readers as the current status, problems, and potential of textile based piezoelectric strain sensor.

After reviewing the revised version of this manuscript, I can now recommend accepting this manuscript for publication in Nature Communications.

[Note from the Editor: Reviewer #2 was asked to look also over the response given to Reviewer #1]

I have investigated all comments from Reviewer #1 and I found some of his/her concerns are not addressed well and they need more clarification or experiments to prove the concept.

6. The authors illustrated that sweating promoted the electrical output of the 3DPF. However, the mechanism interpretation is not scientific. The electrical output of the piezoelectric device is originated from the induced charge rather than electron flow. The faster electron transport of piezoelectric layer would not lead to electrical performance enhancement. In addition, the sweat will be transported to the outer layer and not wet the PVDF layer. Therefore, I can hardly agree with your declaration.

8. The sensitivity should be fitted through a series of data. Three points shown in Figure 4d are not enough to show the relationship between output performance and forces.

These two points need to be addressed with more attention since their answer is not clear. For instant for the question 8, authors should provide sensitivity result for the fabricated samples (V/kPa)

Altogether, I found this manuscript very informative and interesting for publication but revising aforementioned comments are required for final decision.

Reviewer #3 (Remarks to the Author):

I am writing to provide my final assessment of the revised manuscript titled "Development of a 3D Piezoelectric Fabric Sensor for Enhanced User Comfort". Having thoroughly reviewed the authors' responses to my initial feedback, I am pleased to report that the revisions made have significantly strengthened the manuscript.

The authors have addressed each of my comments with careful consideration, providing detailed explanations and additional data where necessary. I would like to highlight the key improvements made in response to the major concerns raised during the initial review:

Originality and Significance:

The authors have successfully emphasized the originality of their work, detailing the unique features of the 3D textile structure piezoelectric sensor. The revised abstract and conclusion now better articulate the distinctive contributions of the research, particularly in terms of comfort, sensitivity, and liquid transport performance.

Minimum Requirements for 3D Automatic Looms:

The manuscript now includes a more elaborate explanation of the minimum requirements for 3D automatic looms and the significance of using hot-stretched yarns. This clarification enhances the reader's understanding of the practical applications of the research.

Numerical Difference in Output Signal:

The authors have addressed the request for numerical differences in output values between the original 3DPF and the dry 3DPF after sweat has evaporated. The provided information on peak volatility adds valuable insights to the results.

Durability Testing and Odor After Washing:

The manuscript now includes comprehensive data on the washability of the 3DPF, meeting the concerns regarding its efficiency after going through laundry. Additionally, the absence of odor after several cycles of sweating and drying has been appropriately addressed.

Application Realism and Concealed Alarm Device:

The authors have provided a more detailed rationale for the choice of the 3DPF belt as a concealed alarm functionality, considering factors such as breathability, comfort, and the concealment of electronics. The versatility of the 3DPF fabric in various applications has been appropriately highlighted.

Grammar and Language:

A professional language editor has been engaged to ensure grammatical accuracy and clarity throughout the manuscript, addressing the concerns raised regarding several grammatical errors.

In light of these revisions, I am confident in recommending the manuscript for publication. The authors have demonstrated a thorough commitment to addressing the reviewers' feedback, resulting in a significantly improved and well-rounded contribution to the field. I appreciate the authors' diligence and responsiveness throughout this process.

Response to reviewers

Manuscript ID: NCOMMS-23-26669

Title: **Sweat permeable and ultrahigh strength 3D PVDF piezoelectric nanoyarn fabric strain sensor**

Dear respected reviewers:

We sincerely thank you for writing us the following constructive comments on our manuscript. Also, we appreciate your willingness to check and help improve the overall contents and quality of our manuscript with your precious time. We have tried our best to revise and improve our manuscript to acknowledge the reviewers' comments accordingly. The comments from the reviewers are *retyped below in italics*. The part with the **blue font** contains the answer point by point. The **red font** in the revised manuscript is the modified content. Thank you so much for your comments and advice.

Reviewers' comments:

Reviewer #2

Authors have done an interesting work to develop a fundamental aspects of fabrication of textile based piezoelectric materials and providing a systematic research on different approaches to maximize the piezoelectric performance of the developed yarn structures using the conjugate electrospinning following by the post treatments. Authors have also mentioned different set up to improve the piezoelectric properties of PVDF based fibers. Moreover, they succeed to provide an image to their potential readers as the current status, problems, and potential of textile based piezoelectric strain sensor.

After reviewing the revised version of this manuscript, I can now recommend accepting this manuscript for publication in Nature Communications.

Answer: We sincerely thank you for recognizing and acknowledging our work.

I have investigated all comments from Reviewer #1 and I found some of his/her concerns are not addressed well and they need more clarification or experiments to prove the concept.

Answer: We have elaborated on two of the reviewers' concerns, the details are as follows.

6. *The authors illustrated that sweating promoted the electrical output of the 3DPF. However, the mechanism interpretation is not scientific. The electrical output of the piezoelectric device is originated from the induced charge rather than electron flow. The faster electron transport of piezoelectric layer would not lead to electrical performance enhancement. In addition, the sweat will be transported to the outer layer and not wet the PVDF layer. Therefore, I can hardly agree with your declaration.*

Answer: We have repeated and verified the experiment results based on the reviewer's suggestion. We have also significantly improved the mechanism analysis to reinforce the accuracy of our findings. To make the answers more targeted, we have broken the above questions into two smaller ones to answer each one

Question 1: The electrical output of the piezoelectric device is originated from the induced charge rather than electron flow. The faster electron transport of piezoelectric layer would not lead to electrical performance enhancement.

Reply: We know that the electrical output of the piezoelectric device is originated from the induced charge rather than electron flow. To ensure the validity and precision of the experiment, relevant tests have been conducted 10 times on five 3DPF samples with the same specifications. The results of the experiment confirmed that both the piezoelectric voltage and current output increased in the wet state. Supplementary Movie 4 shows that the electrical output of the 3DPF after it was absorbed sweat. The finding is consistent with the relevant literatures [*Nano Lett.* 2009 ,**9**, 3720; *Nano Lett.*

2016, **16**, 2400; *J. Am. Chem. Soc.* 2023, **145**, 1936]. Xiong et al. proved that the piezoelectricity of the piezo-material significantly improved by introducing defect carriers [*J. Am. Chem. Soc.* 2023, **145**, 1936]. The sweat is a kind of charge carrier. The insulated PVDF yarn can be considered a piezo-semiconductor material in suitable humidity conditions such as wetting by sweat. Therefore, both theoretically and experimentally, it has been proved that a certain sweat can improve the electrical output of the 3DPF.

As is well known, all piezoelectric materials are dielectric and present an insulating state. However, a suitable moisture amount helps to increase the ferro-/piezoelectric performance of piezoelectric materials [*"Sens. Actuators, A"* 2010, **158**, 106; *"Sens. Actuators, A"* 2011, **167**, 19; *J. Biomechanics* 1976, **9**, 495; *Nature* 1970, **227**; *JCS-Japan* 1994,**102**, 537]. This is because the water molecular adsorption of the piezoelectric materials is a multiple physical and chemical process, which results in the transfer of concomitant electrons from the material [*Nano Lett.* 2009, **9**, 3720; *J. Chem. Phys. C* 2014, **118**, 15910; *J. Am. Chem. Soc.* 2007, **129**, 15684] [*J. Appl. Phys.* 2015, **118**, 15]. For example, the ferroelectric polarity and the piezoelectric performance of piezoelectric materials (e.g., LiNbO₃, BaTiO₃) were changed when it was exposed to liquid water, even sweat [*Nano Lett.* 2009, **9**, 3720; *Nano Lett.* 2016, **16**, 2400]. Another example is that the construction of chemical bonds occurs at the surface of BiFeO₃ in an aqueous solution, which can lead to a change in ferroelectric polarization. This is because enough polarization-selective chemical bonds at the ferroelectric material surface were formed when exposed to the salt ion concentration in the sweat. The ionic introduction of salt ions on the surface can increase the electrostatic energy. At the BaTiO₃-water interface, the adsorption of cation ions from the salt liquid of the sweat along with chemical reactions, can lead to controllable chemical bonds. The large

surface electric potential creates a strong atomic displacement of the Ti atom, causing it to move to the oxygen octahedral structure. As a result, metal electrode-O-H at the BTO piezoelectric sample surface is formed, leading to excellent piezoelectric properties of piezo-semiconductor materials [*Nature* 2014, **514**, 470; *Nat. Rev. Mater.* 2016, **1**, 7; *Nat. Commun.* 2021, **12**, 3508; *Nature* 2022, **608**, 69]. The sweat is a kind of charge carrier, the insulated PVDF nanoyarn is semi-conducted by the introduction of sweat, which in turn enhances the piezoelectric properties of 3DPF.

Question 2: In addition, the sweat will be transported to the outer layer and not wet the PVDF layer.

Reply: The water absorption of the PVDF yarn is weak, but it does not mean that it is completely non-hygroscopic (Figure S1). Wetting was observed in the PVDF layer when the sweat is transmitted from the inner layer to the outer layer of 3DPF through the Z-yarn, but the water absorption was very weak. In addition, the anti-gravity unidirectional liquid transport phenomenon of the 3DPF has been investigated numerically using the COMSOL Multiphysics simulation software. The results show that the droplets pass through the PVDF layer and then go up to the top layer. (Figure S5, Supplementary Movie 3 and Supplementary Movie 4), so the PVDF layer must absorb a certain amount of water. It is because PVDF yarn has weak water absorption and the 3DPF has a strong unidirectional liquid transport ability, so that the PVDF layer absorbs a small amount of water during the water conduction process, resulting in enhanced piezoelectric properties, and on the contrary, if the PVDF layer absorbs a large amount of water, it may lead to reduced piezoelectric properties of the 3DPF.

This water-induced enhancement of ferroelectric polarization [*Nat. Commun* 2018, **9**, 3809] and piezoelectric performance [this work] opens new opportunities for piezoelectric PVDF textile composite in sensing applications.

Figure S1. Comparison the water absorption of Coolmax yarn with polyester yarn, viscose yarn, conductive yarn and PVDF yarns.

Figure S5. The anti-gravity unidirectional liquid transport phenomenon in the 3DPF in (a) isometric view and (b) main view angles.

In the revised manuscript, we have summarized the mechanism of the increase of piezoelectric properties caused by sweat.

In the main text:

(Page 13, Line 19-25) As a result of human sweat, the piezoelectric properties of the 3DPF are not weakened but are enhanced. This is because a suitable moisture amount helps to increase the ferro-/piezoelectric performance of piezoelectric materials⁵¹⁻⁵⁵. In addition, all piezoelectric materials are dielectric and present an insulating state⁵⁶. Piezoelectric materials can improve their piezoelectric properties by introducing defects^{57,58}. The sweat is a kind of charge carrier. Therefore, the insulated PVDF yarn had been semi-conducted by the introduction of sweat defects, which in turn enhances the piezoelectric properties of 3DPF.

8. The sensitivity should be fitted through a series of data. Three points shown in Figure 4d are not enough to show the relationship between output performance and forces. Authors should provide sensitivity result for the fabricated samples (V/kPa)

Answer: Thank you for your suggestion. The number of experiments has been increased to 15 and the relevant data has been added to Fig. 4d in the revised manuscript. Additionally, the sensitivity of the straight-line segment data has been analysed, revealing that the sensitivity of 3DPF is 0.41 V kPa^{-1} in the low-pressure range of 0–1.25 kPa.

Fig.4d. Voltage sensitivity of the 3DPF

We have re-described the sensitivity of 3DPF in the revised manuscript (*line 10, Page 14*).

In the main text:

(*Page 14, Line 9-11*) where v represents the constants. According to Eq. (2), in the low-pressure range of 0–1.25 kPa, the voltage sensitivity values of the 3DPF before and after sweating are 0.41 and 3.95 V kPa⁻¹, respectively (Fig. 4d and Fig. S8).

Reviewer #3

Having thoroughly reviewed the authors' responses to my initial feedback, I am pleased to report that the revisions made have significantly strengthened the manuscript.

The authors have addressed each of my comments with careful consideration, providing detailed explanations and additional data where necessary. I would like to highlight the key improvements made in response to the major concerns raised during the initial review:

Originality and Significance:

The authors have successfully emphasized the originality of their work, detailing the unique features of the 3D textile structure piezoelectric sensor. The revised abstract and conclusion now better articulate the distinctive contributions of the research, particularly in terms of comfort, sensitivity, and liquid transport performance.

Minimum Requirements for 3D Automatic Looms:

The manuscript now includes a more elaborate explanation of the minimum requirements for 3D automatic looms and the significance of using hot-stretched yarns. This clarification enhances the reader's understanding of the practical applications of the research.

Numerical Difference in Output Signal:

The authors have addressed the request for numerical differences in output values between the original 3DPF and the dry 3DPF after sweat has evaporated. The provided information on peak volatility adds valuable insights to the results.

Durability Testing and Odor After Washing:

The manuscript now includes comprehensive data on the washability of the 3DPF, meeting the concerns regarding its efficiency after going through laundry.

Additionally, the absence of odor after several cycles of sweating and drying has been appropriately addressed.

Application Realism and Concealed Alarm Device:

The authors have provided a more detailed rationale for the choice of the 3DPF belt as a concealed alarm functionality, considering factors such as breathability, comfort, and the concealment of electronics. The versatility of the 3DPF fabric in various applications has been appropriately highlighted.

Grammar and Language:

A professional language editor has been engaged to ensure grammatical accuracy and clarity throughout the manuscript, addressing the concerns raised regarding several grammatical errors.

In light of these revisions, I am confident in recommending the manuscript for publication. The authors have demonstrated a thorough commitment to addressing the reviewers' feedback, resulting in a significantly improved and well-rounded contribution to the field. I appreciate the authors' diligence and responsiveness throughout this process.

Answer: We are very grateful for your positive comments on our work.

REVIEWERS' COMMENTS

Reviewer #2 (Remarks to the Author):

After a careful investigation on the revised version of the manuscript I am now confident that this manuscript can be accepted for publication at Nature communication.